# Contrastive Representations for Temporal Reasoning

Alicja Ziarko[1 2 3]     Michał Bortkiewicz[4]     Michał Zawalski[1, 6]
Benjamin Eysenbach[5 †]     Piotr Miłoś[1 3 †*]
[1]University of Warsaw     [2]IDEAS NCBR     [3]IMPAN
[4]Warsaw University of Technology     [5]Princeton University     [6]NVIDIA
aa.ziarko@uw.edu.pl

## Abstract

In classical AI, perception relies on learning state-based representations, while planning — temporal reasoning over action sequences — is typically achieved through search. We study whether such reasoning can instead emerge from representations that capture both perceptual and temporal structure. We show that standard temporal contrastive learning, despite its popularity, often fails to capture temporal structure due to its reliance on spurious features. To address this, we introduce **Contrastive Representations for Temporal Reasoning** (CRTR), a method that uses a negative sampling scheme to provably remove these spurious features and facilitate temporal reasoning. CRTR achieves strong results on domains with complex temporal structure, such as Sokoban and Rubik's Cube. In particular, for the Rubik's Cube, CRTR learns representations that generalize across all initial states and allow it to solve the puzzle using fewer search steps than BestFS — though with longer solutions. To our knowledge, this is the first method that efficiently solves arbitrary Cube states using only learned representations, without relying on an external search algorithm.

Website: https://princeton-rl.github.io/CRTR/

## 1 Introduction

Machine learning has achieved remarkable progress in vision [52], control [21], and language modeling [66, 33]. However, it still falls short on tasks that require structured, combinatorial reasoning. Even relatively simple problems, such as planning in puzzles or verifying symbolic constraints, remain challenging for end-to-end learning systems [61, 41]. The best methods for solving these problems use computationally expensive search algorithms, such as A* or Best First Search (BestFS) [29].

This work centers on the question: *Can we learn representations that reduce or eliminate the need for search in combinatorial reasoning tasks?* To see how good representations can reduce test-time search, consider checking whether a graph has an Euler path. This is true if and only if the graph has exactly zero or two vertices of odd degree. A representation that encodes vertex degrees reduces the task to a simple check—eliminating the need for graph traversal. We approach our question by leveraging temporal contrastive learning [46, 56, 21, 19, 44]. These self-supervised techniques are designed to acquire compact, structured representations that capture the problem's temporal dynamics, enabling efficient planning directly within the latent space.

While (Contrastive Learning) CL has shown promise in control tasks [59, 6], particularly through methods like Contrastive Reinforcement Learning (CRL)[21], we observe that its effectiveness in

---

[*]Now at Google.

[†]Equal advising contribution.

39th Conference on Neural Information Processing Systems (NeurIPS 2025).

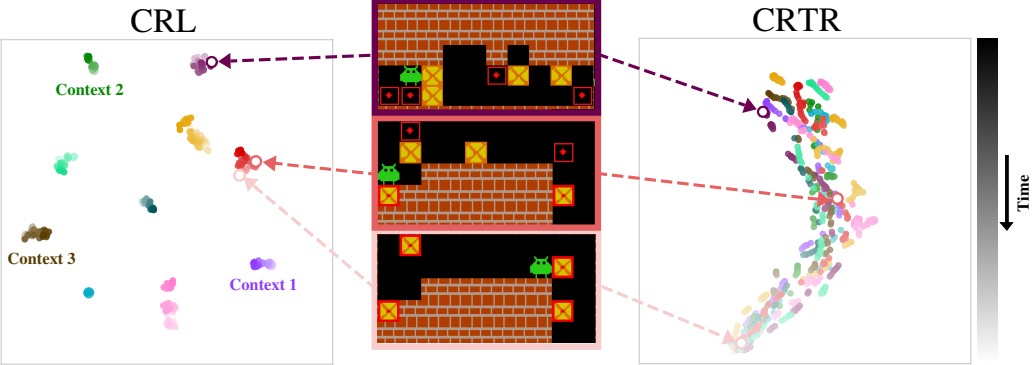

Figure 1: **CRTR makes representations reflect the temporal structure of combinatorial tasks.** t-SNE visualization of representations learned by CRTR (right) and CRL (left) for Sokoban. Colors correspond to trajectories; three frames from two trajectories are shown in the center and linked to their representations. CRL embeddings cluster tightly within trajectories, failing to capture global structure and limiting their usefulness for planning. In contrast, CRTR organizes representations meaningfully across trajectories and time (vertical axis).

combinatorial domains is significantly limited. Specifically, we identify a critical failure mode where contrastive representations overfit to instance-specific context, instead of reflecting environment dynamics. Consequently, models fail to adequately capture the temporal structure that is vital for effective decision-making. This failure mode — such as when the model overfits to wall layouts in Sokoban (Section 4.1) — manifests as a collapse of trajectory representations into small, disconnected clusters, as illustrated in Figure 1 (left).

To solve this, we introduce Contrastive Representations for Temporal Reasoning (CRTR), a simple, theoretically grounded CL method that uses in-trajectory negatives. By design, CRTR forces the model to distinguish between temporally distant states within the *same* episode. This mechanism prevents it from exploiting irrelevant context — such as visual or layout cues — and instead encourages learning temporally meaningful embeddings that reflect the problem's relevant dynamics. This echoes recent findings in neuroscience, where hippocampal representations of overlapping routes diverge during learning despite visual similarity [10]. Our approach similarly prioritizes temporally meaningful structure over reliance on visual cues.

We evaluate CRTR across challenging combinatorial domains: Sokoban, Rubik's Cube, N-Puzzle, Lights Out, and Digit Jumper. Due to their large, discrete state spaces, sparse rewards, and high instance variability, they serve as challenging testbeds for evaluating whether learned representations can support efficient, long-horizon combinatorial reasoning. In each case, CRTR significantly improves planning efficiency over standard contrastive learning, and approaches or surpasses the performance of strong supervised baselines.

Our main contributions are the following:

1. We identify and analyze a critical failure mode in standard contrastive learning, showing its inability to capture relevant temporal or causal structure in problems with a complex temporal structure.

2. We propose Contrastive Representations for Temporal Reasoning (CRTR), a novel and theoretically grounded contrastive learning algorithm that utilizes in-trajectory negative sampling to learn high-quality representations for complex temporal reasoning.

3. We demonstrate that CRTR outperforms existing methods on 4 out of 5 combinatorial reasoning tasks, and that its representations, even without explicit search, enable solving the Rubik's Cube from arbitrary initial states using fewer search steps than BestFS—albeit with longer solutions.

## 2   Related Work

We build upon recent advances in self-supervised RL and contrastive representation learning.

**Contrastive learning.** Contrastive learning is widely used for discovering rich representations from unlabeled data [12, 11, 32, 54] to improve learning downstream tasks [64]. Contrastive learning has enabled effective learning of large-scale models in computer vision [70, 9, 52, 40] and language [60, 24]. Contrastive learning acquires representations by pulling representations of similar data points, i.e., ones that belong to the same underlying concept, closer together and pushing dissimilar ones further apart in the representation space [63]. This idea is reflected in various contrastive objectives, including Triplet Loss [30], NCE [26], InfoNCE [58], and conditional InfoNCE (CCE) [42]. Our work is most closely related to prior work that uses contrastive learning to obtain representations of time series data [46, 56, 21, 3].

**Contrastive representations for sequential problems.** Self-supervised contrastive learning has also been applied to sequential (or temporal) problems, including goal-conditioned RL [21, 62, 44], skill-learning algorithms [48, 73, 18], and exploration methods [25]. Contrastive representations also excel at symbolic reasoning for simple mathematical problems [50]. Most temporal-based contrastive algorithms are based on optimizing the InfoNCE objective [58] to distinguish real future states in the trajectory from random states in other trajectories. Interestingly, Eysenbach et al. [19] demonstrate that intermediate state representations can be effectively approximated via linear interpolation between the initial and final embeddings. Based on these findings, we hypothesize that such representations might also reduce or eliminate the need for search in complex combinatorial problems.

**Combinatorial problems.** Combinatorial environments are characterized by discrete, compact observations that represent exponentially large configuration spaces, often associated with NP-complete problems [35]. Recent RL advancements address these challenges using neural networks to learn efficient strategies, including policy-based heuristics [43, 5], graph neural networks for structural exploitation [8, 36], and imitation learning with expert demonstrations [57]. Our work shows that effective search-facilitating representations can be learned from suboptimal data, without relying on expert demonstrations.

**Planning in latent space.** Efficient planning in complex environments can be achieved by learning state representations that reflect the underlying structure of the problem. Prior work on world models [27, 28] shows that compact representations of high-dimensional environments are critical for agent performance. Another line of research [55, 22] focuses on learning representations that retain only the features relevant for planning. In robotic settings, similar approaches train latent representations to support movement and decision-making [31, 22]. We build on the approach of Eysenbach et al. [21], which formulates goal-conditioned reinforcement learning as a representation learning problem, with the key distinction that we focus on combinatorial reasoning tasks.

## 3 Preliminaries

**Combinatorial problems and dataset properties.** We focus on combinatorial problems, which can be formulated as deterministic goal-conditioned controlled Markov processes $(\mathcal{S}, \mathcal{A}, p, p_0, r_g, \gamma)$. At each timestep $t$, the agent observes both the state $s_t \in \mathcal{S}$ and the goal $g \in \mathcal{S}$, and performs an action $a_t \in \mathcal{A}$. We assume that the transition function $p : \mathcal{A} \times \mathcal{S} \to \mathcal{S}$ is known and deterministic. Initial states are sampled from the distribution $p_0(s_0)$. We define reward function $r_g(s_t) = 1$ for $s_t = g$ and $r_g(s_t) = 0$ otherwise.

The objective is to learn a goal-conditioned policy $\pi(a \mid s, g)$ that maximizes the expected reward: $\max_\pi \mathbb{E}_{p_0(s_0), p_g(g)} \left[ \sum_{t=0}^\infty \gamma^t r_g(s_t, a_t) \right]$. We study an offline learning setup with a dataset of successful yet suboptimal trajectories $\tau_i = ((s_1, a_1), (s_2, a_2), \dots (g, -))$. A state $s_n$ is defined as reachable from $s_1$ if there exist a path $a_1, a_2, \dots, a_n$, such that $s_n = p(a_n, p(a_{n-1}, p(\dots, p(a_1, s_1))))$.

**Contrastive reinforcement learning.** We employ a contrastive reinforcement learning (CRL) method [21] to train a critic $f(s, g)$, which estimates the similarity between the current state $s$ and future state $g$ (goal). The critic uses a single encoder network $\psi$ to generate both representations of the state $\psi(s)$ and the goal $\psi(g)$. Critic's output measures a similarity between these representations with a metric $f_\psi(s, g) = \|\psi(s) - \psi(g)\|$ that reflects the closeness of the states. For details, see Appendix C. To train the critic, we construct a batch $\mathcal{B}$ by sampling $n$ random trajectories from the dataset. For each trajectory, we select a state $s_i$ uniformly and draw a goal $g_i$ using a $\text{Geom}(1 - \gamma)$ distribution over future states. Negative pairs consist of state-goal pairs $(s_i, g_j)$ with goals sampled

from different trajectories. We use the critic's outputs in the InfoNCE objective [58], following prior work in CRL [21, 20, 72, 71, 44, 6]. This objective provides a lower bound on the mutual information between state and goal representations:

$$\min_{\psi} \mathbb{E}_{\mathcal{B}} \left[ - \sum_{i=1}^{|\mathcal{B}|} \log \left( \frac{e^{f_{\psi}(s_i, g_i)}}{\sum_{j=1}^{K} e^{f_{\psi}(s_i, g_j)}} \right) \right].$$

**Mutual information.** The mutual information between random variables $X$ and $Y$ is the amount of information the value of one conveys about the value of the other. Formally, this corresponds to how much the entropy decreases after observing one of the random variables: $I(X; Y) = \mathcal{H}[X] - \mathcal{H}[X \mid Y]$. The **conditional mutual information** $I(X; Y \mid C)$ measures the extra information $Y$ provides about $X$ once the context $C$ is known: $I(X; Y \mid C) = \mathcal{H}[X \mid C] - \mathcal{H}[X \mid Y, C]$. The conditional MI is zero when $X$ and $Y$ are conditionally independent given $C$.

**Search-based planning.** Several prior works use explicit search algorithms solving complex environments [57, 7, 67, 47]. In our study, we focus on the Best-First Search (BestFS) [49]. BestFS builds the search tree by greedily expanding nodes with the highest heuristic estimates, hence targeting paths that are most likely to lead to the goal. In our approach, BestFS uses the representations of the current state and goal, and relies on the critic to evaluate the distances between neighboring states and the goal. While not ensuring optimality, BestFS provides a simple yet effective strategy for navigating complex search spaces. The pseudocode for BestFS is outlined in Appendix B. In our work, we use distances in the latent space as the heuristic, as detailed in Section 3.

## 4   Learning Temporal Representations that Ignore Context

The main contribution of this paper is a method for learning representations that *facilitate planning*. We start by describing how naïve temporal contrastive representations fail in combinatorial problems. Using Sokoban as an example, we will highlight why this happens (Sec. 4.1), and use it to motivate (Sec. 4.1) a different contrastive objective that better facilitates planning on many problems of interest. Section 4.3 summarizes our full method, CRTR.

**Problem definition.** We use a neural network $\phi : \mathcal{S} \mapsto \mathbb{R}^k$ to embed states into $k$-dimensional representations. We define the *critic* $f(s, g) = \|\phi(s) - \phi(g)\|$ as the norm between these learned representations (see Appendix C). Our aim will be to learn effective representations from a dataset of trajectories $(s_t)_{t=1..N}$, collected from either suboptimal expert or random policies. We evaluate these representations by testing whether their distances reflect the structure needed to solve combinatorial tasks like Rubik's Cube or Sokoban.

### 4.1   Failure of Naïve Temporal Contrastive Learning in Combinatorial Domains

A straightforward approach to learning representations $\phi(s)$ is to use temporal contrastive learning [46, 56] (outlined in Sec. 3): positive pairs are sampled from nearby states within the same trajectory, and negatives from different trajectories. However, applying this approach to a common benchmark (Sokoban) reveals an important failure mode. This section introduces this failure mode and its mathematical underpinnings, Sec. 4.2 introduces an idealized algorithm for fixing this failure mode, and Sec. 4.3 turns this idealized algorithm into a practical one that we use for our experiments in Sec. 5.

Sokoban is a puzzle game where an agent must push boxes to target locations in a maze. Each level (or problem instance) is generated with a random wall pattern that is different from one trajectory to another (see Fig. 1). Fig. 1 shows a t-SNE projection of representations learned by temporal contrastive learning on this task. We observe that embeddings from different mazes form tight, isolated clusters. Appendix Fig. 15 shows the same phenomenon happens on the Digit Jumper task. Thus, the representations primarily encode the layout of the walls and not the temporal structure of the task. The reason representations use those features is that they minimize the contrastive objective. Each batch element typically comes from a different maze, so representations that use the wall pattern to detect positive vs negative pairs achieve nearly perfect accuracy. Thus, we will need a different objective to learn representations that primarily focus on temporal structure, and ignore static features (like the walls in Sokoban). To do this, we will first provide a mathematical explanation for this failure mode, which will motivate the new objective in Sec. 4.2.

**A mathematical explanation.** The failure of temporal contrastive learning can be explained with a *context* variable $c$:

**Definition 4.1.** Let $\mathcal{D} = \{\tau_1, \tau_2, \ldots, \tau_n\}$ be a dataset of trajectories, where each trajectory $\tau = (s_1, \ldots, s_T)$ is a sequence of states. Assume that each state can be written as $s_i = (c_i, f_i)$, where $c_i$ is a component that remains constant over the whole trajectory: $c = c_1 = \cdots = c_T$. Then we can write the trajectory as $(c, (f_1, \ldots, f_T))$. We refer to $c$ as the *context* variable, and $(f_1, \ldots, f_T)$ as the *temporal* part of the trajectory.

For instance, in Sokoban, in each trajectory, we can take the context to be the positions of walls and box goals and the temporal part to be the positions of the player and boxes. It is always possible to decompose a trajectory into a context and temporal part by setting $c = ()$, but we will be primarily interested in cases where the context is disjoint from the temporal part:

**Assumption 4.2.** Let $c$ be a context and $(s_1, \ldots, s_T) \sim \mathcal{D}$ a trajectory. Then for any time steps $i < j$, the state $s_j$ and the context $c$ are conditionally independent given $s_i$: $s_j \perp c \mid s_i$.

This Assumption holds for Sokoban, where the context is the wall layout, since the wall layout can be fully determined by each of the states in the trajectory. We will use this assumption to motivate an idealized algorithm in Sec. 4.2, but our practical method in Sec. 4.3 will not require this assumption. Indeed, our experiments (Section 5) show that the approach also works when the context evolves slowly over time (e.g., for the Rubik's Cube).

## 4.2 Learning Representations that Ignore Context: An Idealized Algorithm

Based on this mathematical understanding of the context, we now introduce an idealized alternative method for learning the representations, which makes use of information about the contexts. Sec. 4.3 introduces the practical version of this algorithm, which does not require any *a priori* information about the context.

The key idea in our method is to sample negative pairs $(x, x_-)$ that have the same context, so that the context features are not useful for distinguishing positive and negative pairs; thus, these context features will not be included in the learned repersentations. Specifically, our idealized method works by first sampling the context $c \sim \mathcal{P}(C)$, then positive pairs $(x, x_+)$ from the conditional joint distribution $\mathcal{P}(X, X_+ \mid c)$ and negatives from the marginal conditional distribution $x_-^{(i)} \sim \mathcal{P}(X \mid c)$ for $i \in \{1, \ldots, N-1\}$. The resulting contrastive learning objective is:

$$\max_f \mathcal{L}(f) \triangleq \mathbb{E}_{\substack{c \sim \mathcal{P}(C), (x_j, x_{j+}) \sim \mathcal{P}(X, X_+|C), \\ x_{j-}^i \sim \mathcal{P}(X|c)}} \left[ \frac{1}{N} \sum_{j=1}^N \frac{e^{f(x_j, x_{j+})}}{e^{f(x_j, x_{j+})} + \sum_{k=1}^{N-1} e^{f(x_j, x_{j-}^k)}} \right],$$

$$\max_f \mathcal{L}(f) \triangleq \mathbb{E}_{\substack{(x_j, x_{j+}) \sim \mathcal{P}(X, X_+), \\ x_{j-}^i \sim \mathcal{P}(X)}} \left[ \frac{1}{N} \sum_{j=1}^N \frac{e^{f(x_j, x_{j+})}}{e^{f(x_j, x_{j+})} + \sum_{k=1}^{N-1} e^{f(x_j, x_{j-}^k)}} \right],$$

where $f(\cdot, \cdot)$ is the learned similarity score between state pairs. For example, in Sokoban, this method changes how negative examples are sampled so that they always have the same wall configuration.

Mathematically, this alternative objective is a lower bound on the conditional mutual information $I(X; X_+ \mid C)$ [42], which can be decomposed using the chain rule for mutual information:

$$I(X_+; X \mid C) = I(X_+; C \mid X) + I(X_+; X) - I(X_+; C).$$

Combined with the assumption that $X_+ \perp C \mid X$ (so $I(X_+; C \mid X) = 0$), we see that this new objective is a lower bound on a difference of mutual informations:

$$\mathcal{L}(f) \leq I(X; X_+) - I(X_+; C).$$

The term $I(X; X_+)$ is what is usually maximized by temporal contrastive learning. The second term, $I(X_+; C)$, is akin to adversarial feature learning [38, 16, 65], which aims to learn representations that do not retain certain pieces of information. Our method achieves a similar effect without the need for adversarial optimization. This context-invariance objective parallels mechanisms observed in biological systems, where overlapping episodic memories are actively decorrelated to reduce interference [10].

**Algorithm 1** CRTR performs temporal contrastive learning, but samples negatives in a different way so that representations discard task-irrelevant context, boosting performance (See Fig. 2).

```
# dataset.shape == [num_traj, traj_len, obs_dim]
t0 = np.random.choice(dataset.shape[1], batch_size)
t1 = t0 + np.random.geometric(1 - discount, batch_size)
traj_id = np.random.choice(dataset.shape[0], batch_size)
# 1 new line of code for CRTR (our approach):
traj_id = np.repeat(traj_id[:batch_size // repetition_factor],
                    repetition_factor, axis=0)
batch = (dataset[traj_id, t0], dataset[traj_id, t1])
# further batch processing, the same for CRL and CRTR
```

### 4.3  A Practical Method

While the idealized method in Sec. 4.2 is useful for analysis, the main challenge with practically implementing this method is that the context is not clearly separable from the observation. For example, upon seeing a Sokoban board for the first time, how should one know that blocks are movable (not part of the context) while walls are not (should be part of the context)? This section proposes a practical algorithm that does not require the assumption from Definition 4.1; our experiments in Sec. 5 will demonstrate that this practical method continues to work when this assumption is violated.

The key idea behind our practical method changes how training pairs are sampled for contrastive learning: for each trajectory included in a batch, sample multiple positive pairs. Contrastive learning treats each batch element as forming negative pairs with all other elements in the batch. When multiple positives from the same trajectory are present, some negative pairs will consist of two states from the same trajectory—but likely sampled from different points in time. These within-trajectory negatives are drawn from a different distribution than the corresponding positives, often involving greater temporal separation. As a result, the model is encouraged to focus on temporal distinctions rather than features that are constant throughout a trajectory.

Implementing this idea in practice requires changing just a few lines of code from prior temporal contrastive learning methods, as highlighted in Algorithm 1). The `repetition factor` governs the proportion of such negatives, thereby providing a controllable mechanism to interpolate between the standard and proposed objectives. Using data sampled in this way guarantees that some negative training pairs in each batch come from the same trajectory. We compare with potential alternative approaches in Appendix I.

### 4.4  What if the context is not constant?

While our theoretical analysis required that the context be clearly separable from the observation, the key insight behind our practical method (Sec. 4.3) was to lift this assumption, allowing the method to be applied without any knowledge of the context, even to problems without a constant context (e.g., the Rubik's Cube).

While we empirically test how our method performs on problems without a fixed context in Sec. 5, here we provide some intuition for why the method might be expected to work in such settings. In the Rubik's Cube, for instance, all states are mutually reachable, making it difficult to define the context. Nevertheless, if we focus on the more shuffled portion of the trajectory, simple heuristics like Hamming distance can classify state pairs with 90% accuracy. This suggests that while each move introduces temporal change in some parts of the cube, others remain unchanged, implicitly forming a type of context. As a result, networks may latch onto features that correlate with this pseudo-context rather than true temporal proximity. In Section 5.2 we empirically demonstrate that our method can also improve performance in settings without a constant context.

## 5  Experiments

Our experiments aim to answer the following specific research questions:

1. Does learning representations that ignore context improve performance on combinatorial reasoning problems? (Sec. 5.2)
2. Do learned representations alone suffice for reasoning, or is explicit search essential? (Sec. 5.3)
3. Are representation learning methods that remove context competitive with successful prior methods for combinatorial reasoning? (Sec. 5.2)
4. What is the relative importance of design decisions, such as how the negatives are sampled and the number of in-trajectory negatives? (Sec. 5.4)

## 5.1 Experimental Setup

**Environments.** We evaluate all methods on five challenging combinatorial reasoning tasks: Sokoban [17], Rubik's Cube, N-Puzzle [34], Lights Out [2], and Digit Jumper [4]. Most of these are NP-hard [15, 13, 53] and serve as standard RL benchmarks [1, 51, 69]. *Sokoban* is a grid-based puzzle where an agent pushes boxes to targets while avoiding irreversible states. *Rubik's Cube* requires aligning each face of a 3D cube to a single color. *N-Puzzle* involves sliding tiles within a $4 \times 4$ grid to reach a goal configuration. *Lights Out* is a toggle-based puzzle aiming to switch all cells to an *off* state. *Digit Jumper* is a grid game where each cell indicates the jump length from that position. See Appendix A for full environment details.

**Baselines.** We compare against three baselines. The *contrastive baseline* performs temporal contrastive learning [56, 46, 21], training representations without in-trajectory negatives. We will refer to this baseline as contrastive RL (CRL) [21]. The *supervised baseline* [14, 68] predicts state distances using a value network trained via imitation on demonstrations. Finally, the *DeepCubeA* [1] baseline learns a value function via iterative one-step lookahead. As a lower bound, we also evaluate the performance of representations from a randomly initialized network. For fairness, the CRL and supervised baselines use the same architecture as CRTR.

We will evaluate methods in two settings: with and without search. When we use search, all methods, including DeepCubeA, use BestFS for planning. During tree search, all actions are considered for Rubik's Cube, N-Puzzle, Digit Jumper, and Sokoban; for Lights Out, expansion is limited to the top ten actions ranked by the value function. All the methods avoid loops by only considering states that were not already processed. Further evaluation details are provided in Appendix D. When evaluating methods without search, we greedily find the neighboring states with minimum predicted distance and select the action leading to that state (recall, we assume dynamics are known and deterministic).

**Metrics.** We evaluate two key aspects of representation quality. Spearman's rank **correlation** measures the alignment between representation-space distance and the actual number of steps each state is from the last state in a trajectory. We compute this for each trajectory in the test set, then average the results over 100 trajectories. A high correlation indicates that states closer in time are also closer in the representation space. We demonstrate that this correlation is a good measure of representation quality in Appendix G. **Success rate** at a fixed computational budget measures the fraction of initial states from which a complete solution is found. This metric reflects whether the learned representations support effective planning.

The training details, including hyperparameters, network architectures, and dataset descriptions, are specified in Appendix C. Code to reproduce our experiments is available online: https://github.com/Princeton-RL/CRTR.

## 5.2 Context-Free Representations for Combinatorial Reasoning

We first analyze the representations learned by CRTR visually, using Sokoban as a testbed because it contains obvious context features (the wall positions). Fig. 1 shows the representations learned by CRTR and compares them with those learned by standard temporal contrastive learning (CRL), which differs from our method by not using in-trajectory negatives. We use t-SNE to reduce the representations down to 2D. The CRL baseline learns representations that cluster together, likely focusing on static features – for each trajectory, all observations get encoded to very similar representations. In contrast, the representations from CRTR likely focus on temporal structure – the fact that observations from different trajectories but similar stages of solving get encoded to similar representations indicates that the representations are ignoring context information (which is irrelevant for decision making).

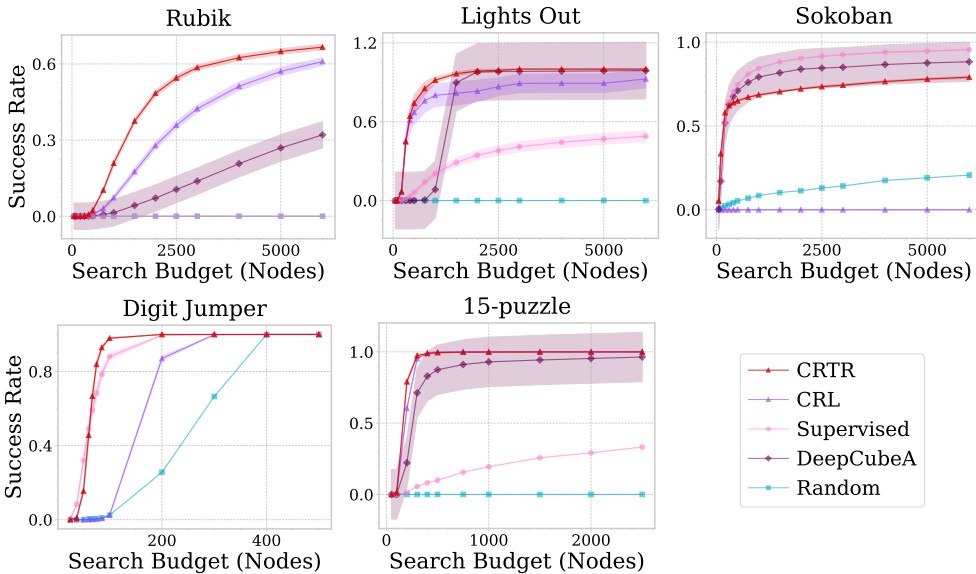

Figure 2: **CRTR performs well in all the evaluated domains.** Success rate as a function of search budget across five domains. CRTR compared to baselines: CRL [21], Supervised [14] and DeepCubeA [1]. Results are averaged over 5 seeds; shaded regions indicate standard error. For the Rubik's Cube, both the supervised and random baselines achieve a success rate of zero for all search budgets.

Our second experiment studies whether these CRTR representations are useful for decision making. To do this, we use the representations to construct a heuristic for search, measuring the fraction of problem configurations that are solved within a given search budget (X axis). As shown in Figure 2, CRTR consistently achieves among the highest success rates, with the largest gains in Sokoban and Digit Jumper. The strong performance relative to CRL highlights the importance of removing context information from learned representations. In Appendix E, we provide additional, smaller-scale experiments showing that these improvements also hold when using a non-greedy solver.

Our third experiment compares CRTR to supervised approaches [1, 14] for solving combinatorial problems. Again, see Fig. 2. CRTR ranks among the top-performing methods in each environment and is strictly the best in two environments. In contrast, the supervised baselines perform much worse in

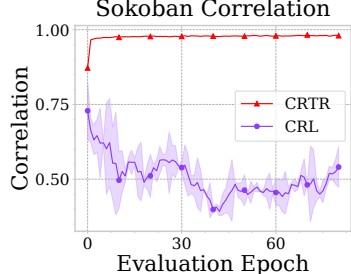

Figure 3: **Distances given by CRTR representations reflect the temporal structure well.** Correlation (Spearman's $\rho$) between the distance induced by learned embeddings and actual distance across the training, CRTR compared with CRL.

Rubik's Cube and Lights Out. We conjecture that the improved performance of CRTR come from how it represents values as distance between learned representations, rather than a number output by a monolithic neural network.

The t-SNE visualizations (Figure 1) suggests that CRL focuses primarily on the static context, while CRTR focuses on the temporal structure. Below, we present additional empirical evidence supporting this interpretation.

We perform further analysis in Sokoban environments. Without negative pairs, the classification task becomes nearly trivial: the model leverages context cues to achieve close to 100% accuracy (Appendix E). Despite this, the learned representations exhibit low correlation with ground-truth state-space distances (Figure 3), indicating that the model ignores temporal structure and instead relies on static context. In contrast, CRTR prevents reliance on contextual shortcuts, resulting in representations that better capture the underlying geometry of the environment (Figure 3). We provide a similar analysis for Digit Jumper in Appendix E. We also demonstrate that using CRTR leads to improved temporal structure in robotic domains (See Appendix F). In Appendix H we show that

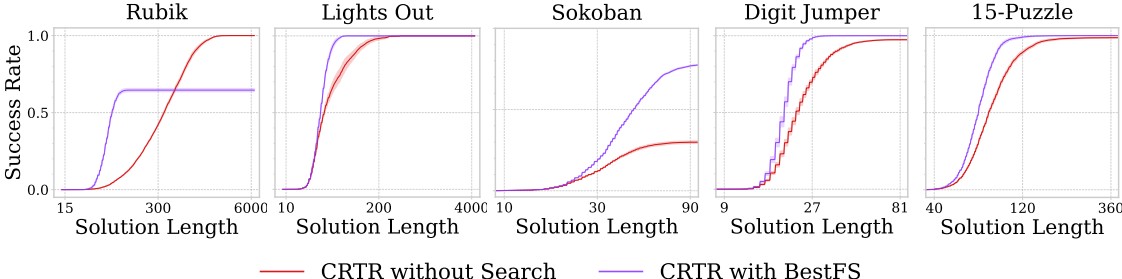

Figure 4: **CRTR solves most tasks without requiring any search.** Adding BestFS results in shorter solutions. We plot the fraction of configurations solved with a solution length of at most $x$, while limiting the number of nodes created to 6000. Surprisingly, on the Rubik's cube CRTR achieves a higher success rate *without* search, solving all board configurations within the budget. Figure 11 in the Appendix E presents the same results, but for CRTR, as well as the Contrastive and Supervised baselines.

CRTR results in representations that optimize conditional mutual information $I(X, X_+|C)$, while CRL does not.

### 5.3 Is search necessary?

Do good representations allow us to solve combinatorial problems without search, or at least reduce the amount of search required to get high success rates? We study this question by using the learned representations to perform greedy planning (see Sec. 5.1) for up to 6000 search steps.

We present the results from this experiment in Figure 4, showing the fraction of problems solved with fewer than a certain number of steps. We compare to the variant of CRTR used in Sec. 5.2 that uses the representations to perform search. As an example of how to interpret this plot, the red line in the leftmost plot shows that CRTR (without search) can solve about 50% of Rubik's cube configurations in less than 300 moves. We compare to the CRL and Supervised baselines in Appendix E.

On 4 / 5 tasks (Rubik's Cube, Lights Out, 15-Puzzle, and Digit Jumper), CRTR solves nearly all problem instances. The key takeaway is thus: *for most problems, CRTR can find solutions without needing any search at all*. Perhaps the most interesting result is the Rubik's cube, where we found that our representations can solve all problem instances in less than 6000 moves. Surprisingly, using search decreases the total fraction of Cube configurations that are solved. The main failure mode is Sokoban, where success rates are low, likely due to the presence of dead-ends in the game.

However, avoiding search comes at a cost: the solutions found without search are typically longer than those found with search. For example, on the Rubik's Cube the average solution length is around 400 moves (see Appendix E) – much higher than the optimal of 26 moves and even higher than the 100 moves that beginner cube solving methods require.

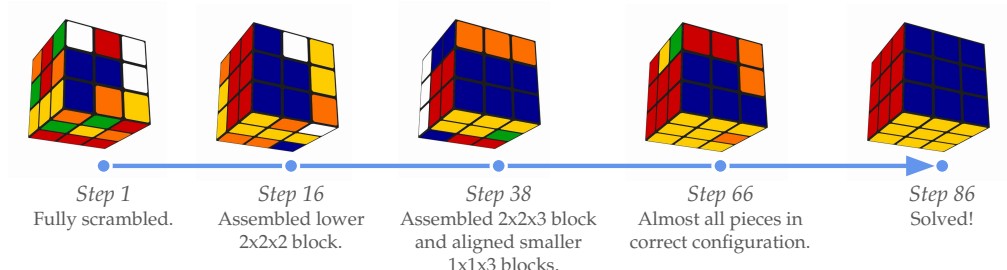

Figure 5: **CRTR without search exhibits a block-building-like behavior.** Intermediate states from solving a randomly scrambled cube, illustrating how the algorithm gradually builds partial structure. The average solve is about 400 moves, and we see similar block building behavior across solves.

This simple greedy approach — just picking the neighbor closest to the goal — starts to show hints of algorithmic behavior. On Rubik's Cube, for example, it learns something that looks like a rudimentary form of block-building (See Fig. 5, a common strategy used by humans for solving the cube. This block building strategy was not programmed or explicitly rewarded, but instead emerged from training the representations on random data.

## 5.4 Additional Experiments

Appendix I presents additional ablation experiments. We find that *(1)* our strategy for sampling data (Alg. 1) outperforms several alternatives, and *(2)* CRTR is robust to the `repetition_factor` hyperparameter, with 2 being a good choice in all settings we have tested.

We report several unsuccessful attempts and negative results in Appendices K and E.

## 6 Conclusion

One way of reading this paper is as a story about search. While much of the early history of AI was built upon search, and while search has regained popularity in recent years as an effective strategy for sampling from language models, our paper shows that many sophisticated reasoning problems can be solved without any search. The key catch is that this is only true if representations are learned appropriately (by removing static context features). While we have only demonstrated results on combinatorial problems, which have known and deterministic dynamics, we look forward to extending these techniques to problems such as chemical retrosynthesis and robotic assembly — problems that have a rich combinatorial structure, but which introduce additional complexity because of unknown and stochastic dynamics.

Another way of reading this paper is as a story about the value of metric embeddings. Image classifiers automatically identify patterns and structures in images (arrays of pixels), mapping images to representations so that semantically-similar images are mapped to similar representations. Our representation learning method does the same for combinatorial reasoning problems, automatically identifying patterns and structures of observations so that the simplest possible decision rule (greedy action selection) can solve some of the most complex reasoning problems (e.g., Sokoban in P-SPACE complete [13]). The success of these representations raises the question of how much reasoning can be done in the representation space. Can we think about compositional reasoning as vector addition? Can we think of length generalization as rays in representation space? And, perhaps most alluringly, can we exploit representational geometry to use the solutions to simple problems to find solutions to hard problems, by using local rules to provide global structure on the space of representations?

**Limitations** Our results characterize the behavior of contrastive methods in deterministic combinatorial environments. However, this work does not account for stochastic settings. Many real-world applications, such as robotic manipulation and autonomous driving, are inherently stochastic, so our findings may not directly generalize to these domains. Moreover, we focus exclusively on fully observable environments with easily identifiable contextual information in the observations. Evaluating whether our approach can be extended to partially observable environments is left for future work.

**Acknowledgments.** The authors would like to thank Raj Ghugare and Mateusz Olko for helpful discussions and feedback. MB was supported by the National Science Centre, Poland (grant no. 2023/51/D/ST6/01609) and the Warsaw University of Technology through the Excellence Initiative: Research University (IDUB) program. MZ was supported by the National Science Centre, Poland (grant no. 2023/49/N/ST6/02819). We gratefully acknowledge Polish high-performance computing infrastructure PLGrid (HPC Center: ACK Cyfronet AGH) for providing computer facilities and support within computational grant no. PLG/2024/017382. We acknowledge Polish high-performance computing infrastructure PLGrid for awarding this project access to the LUMI supercomputer, owned by the EuroHPC Joint Undertaking, hosted by CSC (Finland) and the LUMI consortium through PLL/2024/06/017136.

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

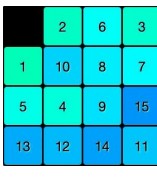
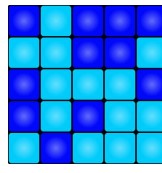
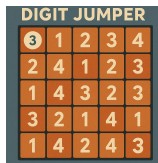

(a) N-Puzzle.          (b) Lights Out.          (c) Digit Jumper.

Figure 6: **Environments:** Our experiments used Sokoban (Fig. 1), the Rubik's Cube (Fig. 5), and the three environments shown above.

## A   Environments

**Sokoban.**   Sokoban is a well-known puzzle game in which a player pushes boxes onto designated goal positions within a confined grid. It is known to be hard from a computational complexity perspective. Solving it requires reasoning over a vast number of possible move sequences, making it a standard benchmark for both classical planning algorithms and modern deep learning approaches [17]. Solving Sokoban requires balancing efficient search with long-term planning. In our experiments, we use 12×12 boards with four boxes.

**Rubik's Cube.**   The Rubik's Cube is a 3D combinatorial puzzle with over $4.3 \times 10^{19}$ possible configurations, making it an ideal testbed for algorithms tackling massive search spaces. Solving the Rubik's Cube requires sophisticated reasoning and planning, as well as the ability to efficiently navigate high-dimensional state spaces. Recent advances in using neural networks for solving this puzzle, such as [1], highlight the potential of deep learning in handling such computationally challenging tasks.

**N-Puzzle.**   N-Puzzle is a sliding-tile puzzle with variants such as the 8-puzzle (3×3 grid), 15-puzzle (4×4 grid), and 24-puzzle (5×5 grid). The objective is to rearrange tiles into a predefined order by sliding them into an empty space. It serves as a classic benchmark for testing the planning and search efficiency of algorithms. The problem's difficulty increases with puzzle size, requiring effective heuristics for solving larger instances.

**Lights Out.**   Lights Out is a single-player game invented in 1995. It is a grid-based game in which each cell (or *light*) can be either on or off. Pressing a cell flips its state and those of its immediate neighbors (above, below, left, and right). Corner and edge lights have fewer neighbors and therefore affect fewer lights. The goal is to press the lights in a strategic order to turn off all the lights on the grid.

**Digit Jumper.**   Digit Jumper is a grid-based game in which the objective is to get from the top-left corner of the board to the bottom-right corner. At each point, the player can move $n$ steps to the left, right, up, or down, where $n$ is determined by the number written on the current cell. *Digit Jumper* is an example of an environment with a constant context, as is *Sokoban*.

## B   Best-First Search

Best-First Search (BestFS) greedily prioritizes node expansions with the highest heuristic estimates, aiming to follow paths that are likely to reach the goal. Although it does not guarantee optimality, BestFS offers a simple and efficient strategy for navigating complex search spaces. The high-level pseudocode for BestFS is presented in Algorithm 2.

---

**Algorithm 2** Best-First Search [29]

---

**while** has nodes to expand **do**
    Take node $N$ with the highest value
    Select children $n_i$ of $N$
    Compute values $v_i$ for the children
    Add $(n_i, v_i)$ to the search tree
**end while**

---

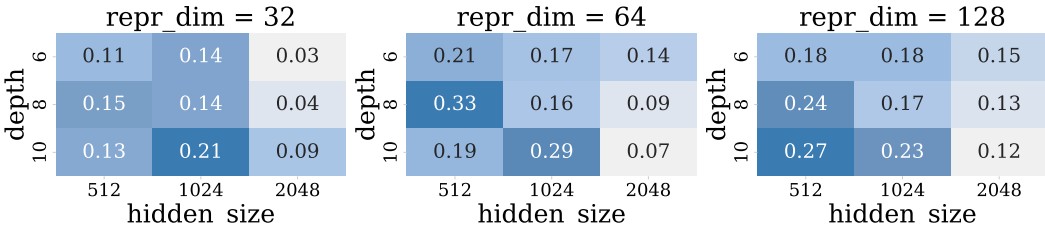

Figure 7: Grid of network's depth, representation dimension and hidden dimension. The success rate is evaluated on cubes scrambled with 10 random moves.

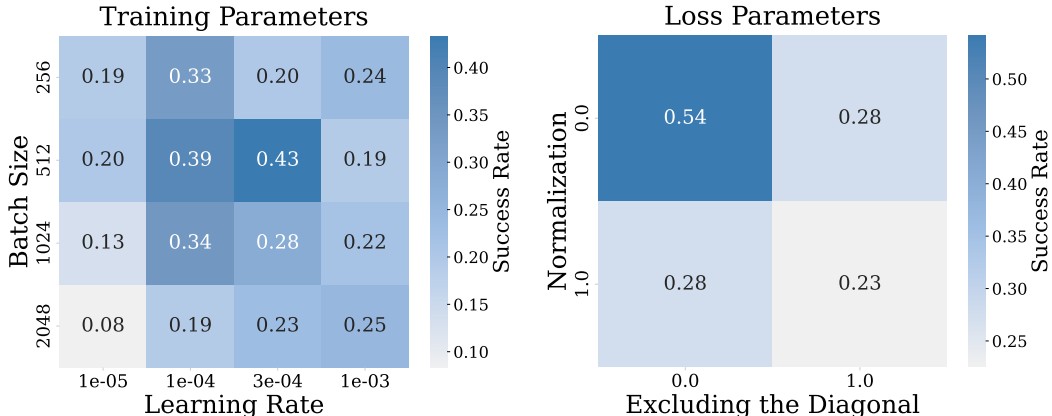

Figure 8: Learning rate and batch size grid for Rubik's Cube. The success rate is evaluated on cubes scrambled with 10 random moves after 700k training steps.

Figure 9: **CRTR is the only effective normalization strategy in Sokoban.** Effect of using negatives in contrastive learning in Sokoban. We compare the setting where the distance to positives is normalized by the sum over all batch elements or only the in-batch negatives. The success rate is evaluated on cubes scrambled with 10 random moves after 400k training steps.

## C   Training Details

Code to reproduce all results is available in the anonymous repository referenced in the main text. Below, we document the training procedures for the supervised baseline, contrastive baseline, and CRTR.

**Training data.**    For Sokoban, we use trajectories provided by Czechowski et al. [14] and train on a dataset of $10^5$ trajectories. For 15-Puzzle, Rubik's Cube, and Lights Out, we generate training trajectories by applying a policy that performs $n$ random actions, where $n$ is set to 150, 21, and 49, respectively. In the case of 15-Puzzle, we additionally remove single-step cycles from the dataset to improve data efficiency. For Digit Jumper, we generate training data by sampling a random path from the upper-left corner to the bottom-right corner on a standard $20 \times 20$ grid. All grid cells not required for this path are filled by sampling uniformly from the set $1, \ldots, 6$. The network for Digit Jumper typically converges after a few hours of training, so we train until convergence is observed. For Sokoban, Rubik's Cube, Lights Out, and 15-Puzzle, we adopt an unlimited data setup and train all models for two days. This results in the models performing approximately $8 \times 10^6$ gradient updates for Rubik's Cube, $7 \times 10^6$ for 15-Puzzle, and $9 \times 10^6$ for Lights Out.

**Training hyperparameters.**    We use the Adam optimizer with a constant learning rate throughout training. A learning rate of 0.0003 was found to perform well across all environments, with the exception of Lights Out, where this setting led to unstable training. For this environment, we instead use a reduced learning rate of 0.0001. In all environments, we use a batch size of 512. The choice of

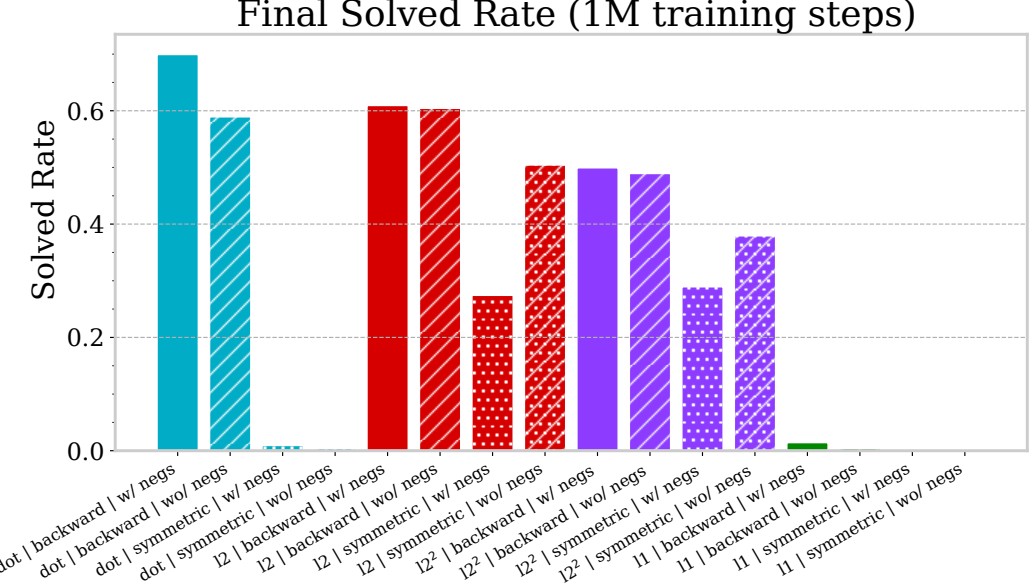

Figure 10: Success rate on Rubik's Cube scrambled with 10 random moves, for models trained with different contrastive losses. Models using the backward loss consistently achieve better performance than those using the symmetric variant. Using the dot product without in-trajectory negatives performs similarly to the $\ell_2$ metric, while combining the dot product with in-trajectory negatives yields the highest success rate. In contrast, combining in-trajectory negatives with symmetric loss results in a drop in performance, likely because, in CRTR, such negatives are often closer to the correct solution in the state-space.

learning rate and batch size was guided by the performance of the contrastive baseline on Rubik's Cube. Specifically, we evaluated solve rates on cubes shuffled 10 times, as shown in Figure 8. We also conducted grid searches to find the optimal training parameters (learning rate and batch size) for the supervised baseline on Sokoban, Lights Out, and Rubik's Cube . We use the same batch size and learning rate across all methods and environments, with the exception of Lights Out, where increasing the batch size and learning rate in the supervised baseline led to a higher success rate.

**Network architecture.** We adopt the network architecture proposed by Nauman et al. [45], using 8 layers with a hidden size of 512 and a representation dimension of 64. This configuration was found to yield optimal performance for the contrastive baseline on Rubik's Cube, as illustrated in Figure 7. We observed that this architecture performs well in all environments except for two cases:

- In Sokoban, a convolutional architecture was required to achieve strong performance.
- In Lights Out, the convolutional network was necessary to ensure training stability.

**Test set.** For Sokoban, we construct a separate test set comprising 100 trajectories, which is used to compute evaluation metrics such as accuracy, correlation, and t-SNE visualizations. For all other environments, a separate test set is unnecessary, as we train for only a single epoch. In this setting, evaluation is performed directly on unseen data sampled during training.

**Contrastive loss.** We use the backward version of the contrastive loss, which we found to consistently outperform the symmetrized variant on Rubik's Cube as shown in Figure 10. We also found the backward version to work better on 15-Puzzle and slightly better in the remaining environments.

For Rubik's Cube, we use the dot product as the similarity metric. Performance across different metrics is presented in Figure 10. While the contrastive baseline performs comparably under the $\ell_2$ metric, CRTR achieves significantly better results with the dot product. Based on similar empirical evaluations, we use the following metrics for other environments:

- Lights Out: $\ell_2$ distance,
- Digit Jumper and 15-Puzzle: dot product,

- Sokoban: squared $\ell_2$ distance.

We set the temperature parameter in the contrastive loss to the square root of the representation dimension.

**Supervised baseline.** The supervised baseline takes as input a pair of states and predicts the distance between them by classifying into discrete bins, where the number of bins corresponds to the maximum trajectory length observed in the dataset.

In all environments, the supervised baseline uses the same architecture as the contrastive baseline.

## D  Evaluation Details

We evaluate all networks on $1000$ problem instances per environment. For Rubik's Cube, each instance is a cube scrambled using $1000$ moves. For 15-Puzzle, Lights Out, and Digit Jumper, evaluation boards are sampled randomly. For Sokoban, we follow the same instance generation procedure as described by Czechowski et al. [14].

## E  Additional Experiments

**A\* solver.** To verify that the improvements achieved by CRTR are not specific to greedy solvers, we conducted an additional experiment using the A\* search algorithm. A\* employs a heuristic function of the form heuristic $+ \alpha \cdot$ cost, where varying $\alpha$ allows trading off between the search budget required to solve the problem and the average solution length. As shown in Table 1, for the Rubik's Cube, increasing $\alpha$ from $0$ (equivalent to BestFS) to $500$ consistently yields better performance for CRTR compared to CRL. We therefore hypothesize that the improvement reported in Section 5.2 is not specific to greedy solvers.

Table 1: **CRTR effectiveness is not BestFS specific.** A\* search results on the Rubik's Cube with a node budget of 6000, varying $\alpha$ in the priority function. CRTR performs better than CRL for all values of $\alpha$, achieving shorter solution lengths and higher solved rates.

| $\alpha$ | 0 | 100 | 200 | 300 | 400 |
|---|---|---|---|---|---|
| CRTR Avg. Solution Length | 56.76 | 46.35 | 38.42 | 32.84 | 29.16 |
| CRTR Success Rate | 0.63 | 0.62 | 0.59 | 0.54 | 0.33 |
| CRL Avg. Solution Length | 62.96 | 49.88 | 41.94 | 36.11 | 31.77 |
| CRL Success Rate | 0.54 | 0.50 | 0.44 | 0.40 | 0.30 |

**No-search results.** The no-search approach selects, at each step, the state that appears most likely to lead toward the solution—based on the learned representation. If the representation were perfect, this strategy would yield optimal solutions. In practice, however, suboptimal representations often cause the agent to wander through latent states far from the goal before eventually converging. As a result, the quality of the representation is reflected in the length of these trajectories: the better it captures directionality in latent space, the shorter the resulting solutions.

Table 2 reports the average solution lengths for the no-search approach on Rubik's Cube and 15-Puzzle. The results suggest that the representations learned by CRTR are better suited to this approach than those learned by the contrastive baseline, and they significantly outperform those derived from the supervised method. This supports the conclusion that CRTR provides a more reliable notion of direction in latent space. Notably, the average solution lengths for both CRTR and CRL are shorter than the length of training trajectories in 15-Puzzle (150), indicating evidence of trajectory stitching.

We furthermore present the distributions of solution lengths for all the methods in Figure 11.

**Accuracy in Sokoban training.** During the training of CRL on the Sokoban environment, a perfect accuracy is acquired almost immediately, due to the method relying on the context, as demostrated in Figure 14.

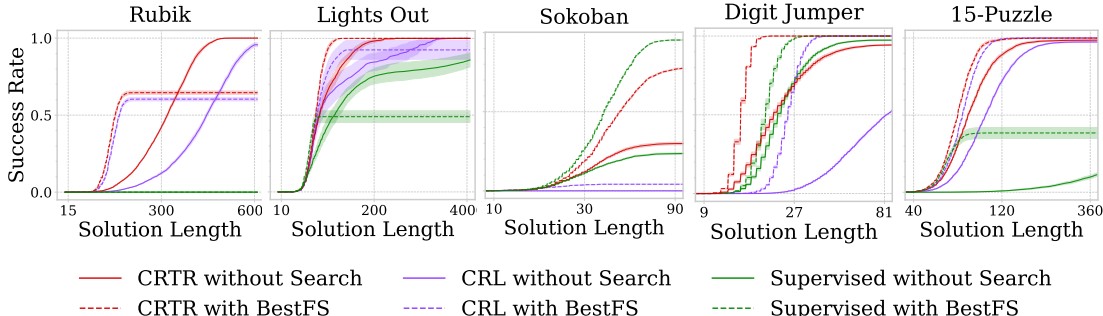

Figure 11: **CRTR produces shorter solutions without explicit search in comparison to baselines. Search can help reduce solution length further.** Fraction of boards solved with a solution length of at most $x$, comparing CRTR to baselines. Figure 4 in the main text presents analogous results, but only CRTR, for clarity.

Table 2: Average solution length of the baselines and CRTR on Rubik's Cube and 15-Puzzle without using search. Supervised baseline fails to solve Rubik's Cube without search.

| Problem | CRTR | Contrastive Baseline | Supervised Baseline |
|---|---|---|---|
| Rubik's Cube | 448.7 | 1830.3 | NaN |
| 15-puzzle | 82.4 | 119.5 | 1054.3 |

**Digit Jumper analysis.** Digit-Jumper is an example of another constant context (defined in Sec. 4.1) environment, as is Sokoban. It is therefore another environment in which CRL fails rather spectacularly and therefore, we observe a similar effect to that seen in Sokoban when comparing CRTR to standard CRL. As shown in Figure 12, CRL rapidly achieves 100% training accuracy. However, despite this perfect accuracy, the resulting representations exhibit poor correlation with actual temporal structure (Figure 13). This is consistent with the t-SNE visualization (Figure 15): as with Sokoban, CRL collapses each trajectory into a single point in the representation space, discarding temporal information. In contrast, CRTR preserves a clear temporal structure within the latent space (see Figure 15). For non-constant context environments, the difference in representation quality is also visible in success rates, accuracy and correlation, it is however much less pronounced.

**Results on SAT.** To compare against GNN baselines, we conducted preliminary experiments evaluating CRTR on the 3-SAT task from G4SATBENCH [39] (easy split). When trained on this benchmark, CRTR was unable to achieve nonzero test performance, whereas GNN-based methods reach nearly 90%. We attribute this discrepancy to the extremely low-data regime: networks in this benchmark are evaluated after training on only 800 samples—orders of magnitude fewer than those used to train CRTR on tasks where it excelled. Supporting this interpretation, we found that simplifying the evaluation by restricting clauses to at most five independent variables (instead of 10–40) substantially improved results. Under this setting, performance increased from random level (15%) to 52% after 5000 training steps, when running the evaluation with the no-search approach. We therefore hypothesize that CRTR is less sample efficient than GNN-based methods, suggesting that further work is needed to improve data efficiency in representation-learning approaches.

**Results on Sudoku.** Given Sudoku's recent popularity as a benchmark for evaluating reasoning in large language models, we also tested our method on this task. CRTR was unable to achieve performance above random chance. We attribute this to the goal-conditioned nature of our approach: solving a Sudoku puzzle requires reasoning toward an unknown final configuration, whereas our method assumes access to a well-defined goal state during inference. While this assumption holds for many of the tasks on which CRTR performs well, it is violated in Sudoku. To mitigate this, we experimented with adding an artificial "solved state" shared across all trajectories, but this did not substantially improve results. We view this limitation as a promising direction for future work on extending goal-conditioned architectures to problems with implicit or unknown goals.

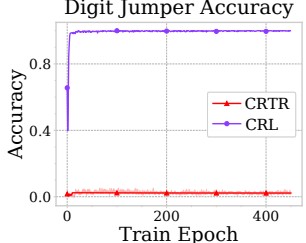

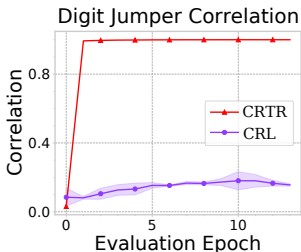

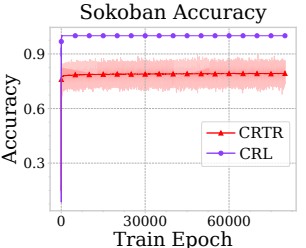

Figure 12: **In Digit Jumper, CRL quickly acquires near-perfect accuracy, however this is due to relying only on superficial features – the board layout.** Accuracy of classifying whether two states form a positive pair across the training, CRTR compared with CRL.

Figure 13: **In Digit Jumper, CRTR improves temporal structure in robotics environments.** Comparison of Spearman's rank correlation metric for $CR^2$ (solid) and CRL (dashed) for D4RL offline datasets.

Figure 14: **In Sokoban, CRL quickly acquires near-perfect accuracy, however this is due to relying only on superficial features, such as walls.** Accuracy of classifying whether two states form a positive pair across training: CRTR compared with CRL. The accuracy saturates at a value smaller than 1 for CRTR, as a result of containing in-trajectory negatives.

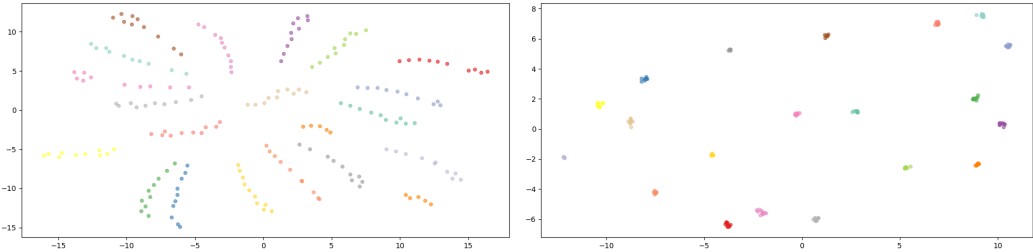

Figure 15: **CRTR makes representations reflect the structure of the combinatorial task.** t-SNE visualization of representations learned by CRTR (left) and CRL (right) for Digit Jumper. Colors correspond to trajectories. CRL representations (right) cluster within trajectories, making them useless for planning.

## F  Generalization to Temporal Reasoning in Non-Combinatorial Domains

To investigate whether CRTR also identifies temporal features in non-combinatorial domains, we apply it to a dataset of robotic manipulation trajectories (the Adroit dataset from D4RL [23]). Those tasks require using a high-dimensional robotic hand to perform fine motor activities, and are designed to test fine motor control and long-horizon planning. We quantify representation quality by measuring the predicted distance from each state in a trajectory to the final state in a trajectory. Specifically, we look at the rank correlation between the time step and predicted distance, with a correlation of 1 indicating that the learned representations are highly predictive of the temporal distance from each state to the final state.

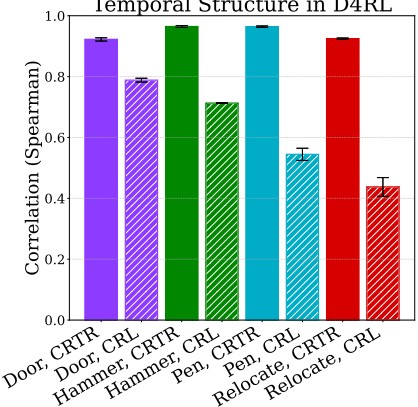

Figure 16: **CRTR improves temporal structure in robotics environments.** Comparison of Spearman's rank correlation metric for $CR^2$ (solid) and CRL (dashed) for D4RL offline datasets.

We look at the correlation through training for CRTR and CRL (Fig. 16). CRTR results in a higher correlation (more than 0.9 in comparison to 0.5 – 0.8 depending on the environment), as well as visibly better training stability – for standard CRL, the correlation is visibly unstable through training and in some cases even becomes smaller as the training progresses. This result is a little surprising, and it is not fully clear why does the improvement happen. We hypothesize that this is because the initial position of the robot differs between trajectories and serves as a sort of slowly

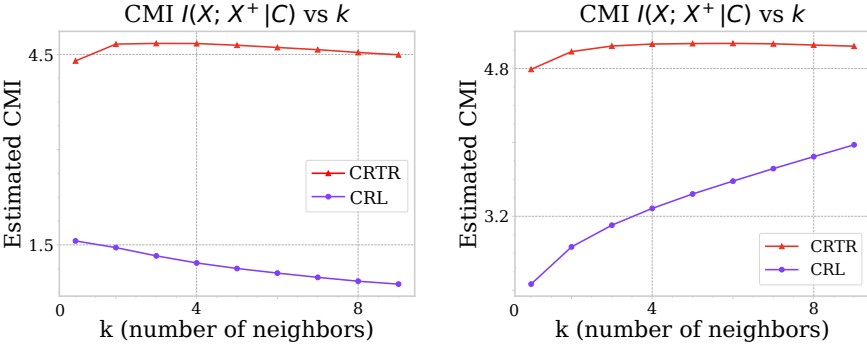

Figure 17: **CRTR optimizes the conditional mutual information while CRL does not, confirming out theoretical results 4.1.** Conditional mutual information estimated in Sokoban (left) and Digit Jumper (right) for representations learned by CRTR and CRL, for different values of nearest neighbors used for estimation.

changing context, similarly to the Rubik's Cube case. We conclude that using CRTR results in a better temporal structure in the representation space for non-combinatorial problems.

## G    Correlation as a Measure of Representation Quality

To assess whether Spearman rank correlation is a reliable indicator of representation quality, we performed a grid of 96 short runs for each of three environments: Sokoban (12×12), Sokoban (16×16), and the Rubik's Cube. We varied four factors: network depth (8, 6, 4, 2), network width (1024, 16), representation dimension (64, 32, 16, 8), and the distance metric used in the contrastive loss (dot product, $\ell_2$, $\ell_2^2$).

Across all environments, the final Spearman correlation (computed with a budget of 1000 nodes) showed a strong relationship with the final success rate: 0.89 for 12×12 Sokoban, 0.80 for 16×16 Sokoban, and 0.90 for the Rubik's Cube. These results support the conclusion that Spearman rank correlation is a good measure of representation quality.

## H    Mutual Information Analysis

To estimate the conditional mutual information, we use NPEET package, which implements the method proposed in [37] that uses k-nearest neighbours for entropy estimation. We conduct the analysis using trajectories collected from the Sokoban or Digit Jumper environment, utilizing all transitions within these trajectories ($> 45k$ transitions for Sokoban and $> 20k$ for Digit Jumper). The variables used in the experiment are defined as follows:

- $X$: Current state embeddings, standardized using z-score normalization (mean 0, standard deviation 1) across the dataset. These embeddings are then projected onto a 3-dimensional subspace using Principal Component Analysis (PCA).

- $X^+$: Next state embeddings corresponding to transitions from $X$. The same standardization parameters and PCA transformation applied to $X$ are used for $X^+$ to ensure consistency.

- $C$: Trajectory identifiers (`traj_id`) encoded as 2-dimensional vectors sampled from a standard bivariate Gaussian distribution (i.e., $\mathcal{N}(0, I_2)$).

To mitigate the effects of the curse of dimensionality and ensure reliable performance of k-nearest neighbor (kNN)-based estimators, we reduce all high-dimensional representations to low-dimensional spaces (3D for state embeddings, 2D for trajectory identifiers). The conditional mutual information for CRTR and contrastive baseline is reported in Figure 17.

## I    Ablations

**Repetition factor.** Our method introduces a single additional hyperparameter: the repetition factor $R$. This parameter controls the proportion of in-trajectory negatives and is critical for achieving strong performance. As shown in Figure 18, the impact of increasing $R$ varies by environment. For Sokoban, higher values of $R$ lead to only a slight decline in performance. In contrast, in many other environments, excessive repetition can significantly degrade results. While $R = 2$ is not always optimal, it consistently improves performance across all environments we evaluated and serves as a strong default choice.

In Figure 19, we present detailed results showing how varying the repetition factor influences the success rate.

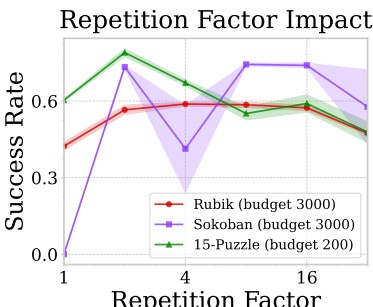

Figure 18: **A repetition factor of 2 consistently improves the performance.** Increasing the repetition factor for Sokoban, N-Puzzle, and Rubik's Cube, respectively.

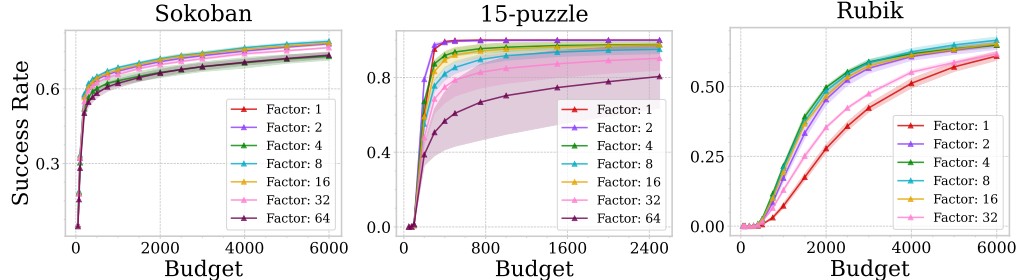

Figure 19: **Influence of the repetition factor depends on the environment type.** Increasing the repetition factor for Sokoban, N-Puzzle, and Rubik's Cube, respectively.

**Negatives.** We explored alternative methods for incorporating in-trajectory negatives into the contrastive loss. The first approach mimics the standard addition of hard negatives: given a batch $\mathcal{B} = (x_i, x_{i+})_{i \in \{1..B\}}$, we sample additional negatives $(x_{i-})_{i \in \{1..B\}}$, and compute the loss as

$$\mathcal{L} = \frac{1}{B} \sum_i \log \left( \frac{\exp\left(f(x_i, x_{i+})\right)}{\sum_{j \neq i} \exp(f(x_i, x_{j+})) + \exp(f(x_i, x_{i-}))} \right).$$

We considered three strategies for selecting in-trajectory negatives: sampling a state uniformly at random, choosing the first state, or choosing the last state of the trajectory. For Rubik's Cube, instead of choosing the last state—which is identical for all trajectories—we sample a random state farther from the solution to serve as a negative.

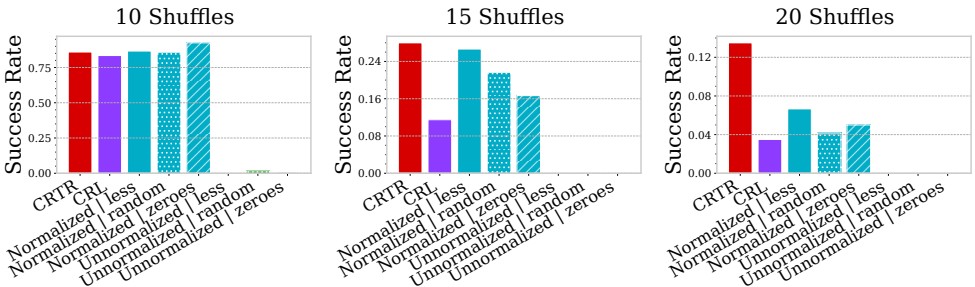

Figure 20: **In Rubik's Cube, CRTR outperforms all negative sampling strategies, when the number of scrambles increases.** Comparison of different methods for introducing in-trajectory negatives in the Rubik's Cube environment, with an increasing number of cube scrambles. While normalized negatives perform similarly to CRTR for a small number of scrambles, their performance deteriorates as the number of scrambles increases.

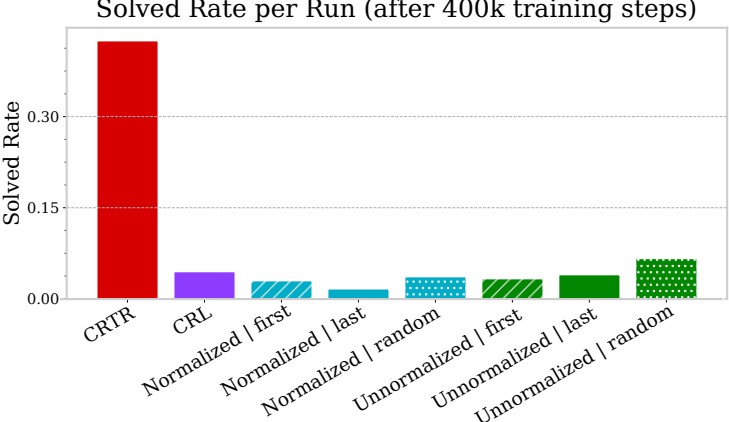

Figure 21: We compare various methods for introducing in-trajectory negatives in the Sokoban environment and find that only CRTR yields effective results.

As shown in Figures 20 and 21, training with this approach did not yield strong performance. We hypothesized that the large prediction error introduced by the in-trajectory negatives $(x_{i-})$ caused excessively large gradients, destabilizing training. To mitigate this, we applied a normalization scheme: ensuring that the vector $[f(x_1, x_{1-}) \quad \cdots \quad f(x_B, x_{B-})]$ has the same Frobenius norm as the $B \times B$ matrix

$$\begin{bmatrix} f(x_1, x_{1+}) & f(x_1, x_{2+}) & \dots & f(x_1, x_{B+}) \\ \vdots & \vdots & \ddots & \vdots \\ f(x_B, x_{1+}) & f(x_B, x_{2+}) & \dots & f(x_B, x_{B+}) \end{bmatrix}.$$

This normalization enabled achieving comparable performance to CRTR on Rubik's Cube scrambled 10 times (Figure 20). However, CRTR still outperforms all negative sampling strategies on cubes scrambled 15 and 20 times.

For Sokoban, the only approach that consistently improved performance is CRTR, as demonstrated in Figure 21. We hypothesize that this is because removing contextual information is more challenging in Sokoban than in Rubik's Cube. In the latter, the context is more local and changes gradually over time, making it *softer*, while the context in Sokoban is constant throughout a trajectory. This is discussed in detail in Section 4.1.

While at first glance, repeating trajectories in a batch may seem equivalent to sampling in-trajectory hard negatives, the two approaches are different. In standard contrastive learning (as in CRL), an anchor $x$ pulls its positive $x_+$ closer and pushes negatives (e.g., $y_+$) away. However, negatives like $y_+$ are simultaneously pulled by their own anchors (e.g., $y$), which limits how far they are pushed by $x$. In contrast, when using in-trajectory negatives without anchoring them (e.g., $x$ pushes $x_-$ away, but $x_-$ has no anchor), these states can drift arbitrarily far in representation space. This is problematic, especially since in-trajectory negatives are harder (closer in structure), which results in stronger gradient updates. Our proposed method, CRTR, addresses this by anchoring all in-trajectory negatives. This keeps trajectories coherent and prevents such drift.

## J   Computational Resources

All training experiments were conducted using NVIDIA A100 GPUs and took between 5 and 48 hours each. The solving runs ranged from 10 minutes to 10 hours. In total, the project required approximately 30,000 GPU hours to complete.

## K   Things We Tried That Did Not Work

- Using separate encoders for future and present states did not improve performance.

- Adding extra layers to encode the action led to lower success rates.

- Using only in-trajectory negatives degraded performance.

- Modifying how current states are sampled in CRL (e.g., deviating from uniform sampling) did not yield improvements.

- Using $A^*$ solver with our representations could be greatly improved. Because distances in the latent space are only monotonically correlated—not linearly correlated—with actual distances, a modification to $A^*$ that would account for these discrepancies could bring huge gains.

- Distances between Rubik's Cube states, measured by the number of actions, almost always satisfy the triangle inequality with equality. Consequently, this metric cannot be faithfully embedded in Euclidean space, where equality in the triangle inequality occurs only for collinear points.

- Since Rubik's Cube actions are not commutative, a faithful Cayley graph structure could only emerge in a Euclidean space where vector addition is noncommutative—which would require a highly non-standard space.

