# OpenReview forum: "Contrastive Representations for Temporal Reasoning"
_NeurIPS.cc/2025/Conference — NeurIPS 2025 poster_

### Official Review · Reviewer_BU6x · 2025-07-02

**Clarity:** 3
**Significance:** 3
**Originality:** 2
**Rating:** 4
**Confidence:** 3

**Summary:**

This paper introduces CR² (Contrastive Representations for Combinatorial Reasoning) to address limitations of standard contrastive learning (CRL) when applied to combinatorial problems. The authors identify that CRL fails in combinatorial domains because it learns to exploit superficial context features (like wall layouts in Sokoban) rather than temporal structure. CR² addresses this by incorporating "in-trajectory negatives" - negative samples from the same trajectory rather than just across different trajectories. By including negatives that have similar context (similar wall layouts), the model is forced to learn temporal structure instead of just memorizing context. This is done by repeating trajectories in the batch, so there is a higher chance that negatives come from trajectories with similar context.


Main Claims of Contributions:

- Failure Mode Identification: Standard CRL fails on combinatorial problems by learning context-dependent representations that ignore temporal structure
- Novel Algorithm: CR² optimizes I(X; X⁺) - I(X⁺; C), encouraging representations that maximize temporal mutual information while minimizing context dependence

**Questions:**

Questions:

- The "repetition factor" hack seems unprincipled - why this specific approach?
- All experiments use BestFS. How do the learned representations perform with other search algorithms (A*, MCTS, etc.)? Is the improvement specific to greedy search strategies?
- The definition of context (Definition 4.1) works well for Sokoban where static elements are clearly separable, but how would you formally define context in domains like Rubik's Cube where you mention "pseudo-context"? Could you provide a more general mathematical framework for identifying context across different combinatorial domains?
- The conditional independence assumption is crucial to your theoretical analysis. Have you empirically validated this assumption across your test domains? What happens to your theoretical guarantees when this assumption is violated?

**Ethical Concerns:**

["NO or VERY MINOR ethics concerns only"]

**Final Justification:**

I appreciate the author response and have read all the other reviews. Most of my gaps in understanding and concerns have been answered. I am able to appreciate the novelty regarding leveraging trajectory structure to construct informative negatives when using contrastive learning to solve combinatorial problems. The new experiments and analysis strengthen the paper. I have increased the score.

**Limitations:**

Yes

**Paper Formatting Concerns:**

No formatting concerns

**Quality:**

2

**Strengths And Weaknesses:**

Strengths:

- The paper addresses an important problem in representation learning: capturing temporal dependencies to enable better planning and improve sample efficiency. The combination of contrastive learning with novel approaches shows promise for learning effective representations.

Weaknesses:

This work requires substantial development (sufficient novelty and additional contributions) before publication. The main issues are:


- The core insight about negative sample selection is reasonable but offers minimal novelty. The proposed solution appears to be an ad-hoc fix rather than a principled algorithm, with weak connections between the theoretical objective and actual implementation.

- The method makes strong, unrealistic assumptions. It only works if we can cleanly separate "context" from "temporal dynamics" - a strong assumption that breaks down in many domains. The paper provides no systematic approach for identifying context in practice.

- Inadequate Experimental Validation: The evaluation doesn't support the broad claims made. Despite the title suggesting general "combinatorial reasoning," testing is limited to puzzle games with no evidence of effectiveness on real combinatorial problems (traveling salesman, vehicle routing, etc.). Additional issues include: No results across multiple random seeds, insufficient analysis of learned representations beyond basic correlation, and missing comparisons to state-of-the-art methods (Graph Neural Networks, neural combinatorial optimization, Monte Carlo Tree Search variants).


- Fig. 3 highlights the difference in the code for CRL and CR2 (which is unusual to add in the main paper). This again suggests an engineering modification rather than a research contribution.

- The related work section is superficial and fails to clearly position the work within existing literature.

Overall Assessment:

The work appears more like an engineering hack than a novel research contribution and needs significant theoretical development and experimental validation before being ready for publication.

---

> ### Author Rebuttal · Authors · 2025-07-31
>
> We thank the reviewer for the detailed feedback, which we believe will strengthen the paper.
>
> **> All experiments use BestFS. How do the learned representations perform with other search algorithms (A\*, MCTS, etc.)? Is the improvement specific to greedy search strategies?**
>
> Following the reviewer’s comment, we ran preliminary A* experiments on the Rubik’s Cube to test whether CR$^2$ representations remain effective beyond greedy search. Using a node budget of 6000 and varying $\alpha$ in the priority function $heuristic + \alpha\cdot cost$ for $\alpha \in {0, \dots, 400}$, CR$^2$ consistently outperforms CRL in both success rate and solution length. This suggests that CR$^2$ generalizes well across search strategies. We will clarify this point and include the results in the revised paper. The full results are in the table below.
> | Coefficient              	| 0 	| 100   | 200   | 300   | 400   |
> |-----------------------------|-------|-------|-------|-------|-------|
> | **CR$^2$ Avg. Solution Length** | 56.76 | 46.35 | 38.42 | 32.84 | 29.16 |
> | **CR$^2$ Success Rate**     	| 0.63  | 0.62  | 0.59  | 0.54  | 0.33  |
> | **CRL Avg. Solution Length**	| 62.96 | 49.88 | 41.94 | 36.11 | 31.77 |
> | **CRL Success Rate**       	| 0.54  | 0.50  | 0.44  | 0.40  | 0.30  |
>
> **> The "repetition factor" hack seems unprincipled - why this specific approach?**
>
> Thank you for raising this point. While the repetition factor may appear ad hoc at first glance, it is in fact motivated both empirically and conceptually.
>
> We tested several natural alternatives for incorporating in-trajectory negatives, and found that our method consistently outperforms them — as shown in Appendix Figures 23 and 24, in both Sokoban and Rubik’s Cube. These comparisons demonstrate that the repetition factor is not just a heuristic convenience, but an empirically validated design choice.
>
> Conceptually, the repetition factor ensures that all in-trajectory negatives are properly anchored. In standard contrastive learning (e.g., CRL), negatives (like $y_+$) are simultaneously pulled by their own anchors (e.g., $y$), limiting how far another anchor (e.g., $x$) can push them. Without anchoring, harder in-trajectory negatives (e.g., $x_-$) can drift arbitrarily far due to stronger gradients. Our approach mitigates this by ensuring that all such states appear as anchors in the batch, which stabilizes training and preserves the structure of individual trajectories.
> We have updated the Method section to clarify this motivation.
>
> **> The conditional independence assumption is crucial to your theoretical analysis. Have you empirically validated this assumption across your test domains? What happens to your theoretical guarantees when this assumption is violated?**
>
> In the constant context setting, if the environment is fully observable and the context $C$ is deterministically derived from the full state $X$, then the conditional independence assumption holds: observing $X^+$ adds no new information about $C$ beyond $X$. We will clarify this in the revised paper.
> More generally, the assumption’s validity depends on how $C$ is defined. While it can be trivially satisfied (e.g., $C$ is empty), this is not meaningful in practice.
> If the assumption is violated, the theoretical guarantees no longer strictly hold; however, our experimental results suggest the method remains practically useful. We have updated the Limitations section to reflect this.
>
> **> Despite the title suggesting general "combinatorial reasoning," testing is limited to puzzle games with no evidence of effectiveness on real combinatorial problems (traveling salesman, vehicle routing, etc.).**
>
> Thank you for this suggestion. Our paper focuses on complex, structured reasoning tasks under deterministic dynamics, as in standard benchmarks like the Rubik’s Cube and 15-puzzle, which are widely used to evaluate planning [2, 3, 4] due to their combinatorial complexity.
> We understand the concern and are open to adjusting the title to better reflect this scope (e.g., Contrastive Representations for Complex Planning Tasks), and would make corresponding edits throughout the paper if preferred.
>
> **> No results across multiple random seeds,**
>
> Thank you for pointing this out. Our main results (Figure 4) were averaged over 3 random seeds, though this was not clearly stated in the main text. In some cases, the variance is very low, making the error bars barely visible without zooming in. We have updated the caption of Figure 4 to clarify this.
> To increase confidence in the results, we are adding more seeds (up to 5) for selected experiments, including those in Figures 4, 8, and 9.
>
> **> insufficient analysis of learned representations beyond basic correlation**
>
> We thank the reviewer for this comment. To assess whether contextual information is present in the representations, we trained a linear classifier to predict trajectory ID, used as a proxy for static context, from representations produced by CRL, CR$^2$, or a randomly initialized network.
>
> We used >49k Sokoban transitions, with 70% of each trajectory for training and 30% for testing, and report mean accuracy over 6-fold cross-validation.
>
> Results (mean ± std):
> - CRL: 88.16 ± 0.30%,
> - CR$^2$: 1.80 ± 0.09%,
> - Randomly initialized representations: 3.32 ± 0.15%.
>
> These results show that CRL retains contextual information, while CR$^2$ effectively suppresses it.
> We have added a brief summary in Section 4.2 and included full details in the Appendix.
>
> **> missing comparisons to state-of-the-art methods (Graph Neural Networks, neural combinatorial optimization, Monte Carlo Tree Search variants).**
>
> Thank you for the suggestion. To the best of our knowledge, state-of-the-art methods such as GNN-based models, neural combinatorial optimization approaches, and MCTS variants have not been applied to the specific puzzle domains we evaluate (e.g., Rubik’s Cube, 15-puzzle, Sokoban) in a way that would allow for direct comparison. If the reviewer is aware of specific methods or references relevant to these domains, we would be grateful for the pointers and would be happy to include them in the paper and, if feasible, in future evaluations.
>
> **> The method makes strong, unrealistic assumptions. It only works if we can cleanly separate "context" from "temporal dynamics" - a strong assumption that breaks down in many domains. The paper provides no systematic approach for identifying context in practice.**
>
> We believe there may be some misunderstanding here: Fig. 4 shows experimentally that the method works when the context is not clearly separable. The point of our practical implementation is that it bypasses the need for identifying the context.
> We agree that the theoretical analysis requires some assumptions (as is common in prior work). We will revise the paper to highlight that these assumptions are not required by our practical method.
>
> **> The related work section is superficial and fails to clearly position the work within existing literature.**
>
> We have revised the related work section to more clearly highlight the similarities and differences from prior work.
> If the reviewer could point to specific areas or references we may have overlooked, we would be happy to update the related work section in the paper to better reflect prior work and clarify our contributions.
>
>
> **> The definition of context works well for Sokoban where static elements are clearly separable, but how would you formally define context in domains like Rubik's Cube? Could you provide a more general mathematical framework for identifying context across different combinatorial domains?**
>
> Thank you for this insightful question. Our formal definition (Definition 4.1) assumes context corresponds to a clearly separable or static part of the state, as in Sokoban.
>
> In domains like Rubik’s Cube, where no explicit static structure exists, we treat “pseudo-context” as a latent variable implicitly shared across a trajectory—e.g., abstract features such as the initial configuration or symmetry class. Our empirical approach approximates conditional sampling using in-trajectory negatives.
>
> Developing a general mathematical framework for identifying or learning context in such combinatorial domains is an important direction for future work. We will clarify this in the paper.
>
> **> The core insight about negative sample selection is reasonable but offers minimal novelty.**
>
> We appreciate the reviewer’s recognition that our core insight on negative sample selection is reasonable. While hard negatives are well-established in contrastive learning, our contribution lies in reframing this idea for a setting where contrastive methods typically struggle—combinatorial puzzles.
>
> We propose a way to leverage trajectory structure to construct informative negatives, enabling success on tasks where prior contrastive methods fail. To our knowledge, this framing and application have not been explored before. If the reviewer is aware of related work, we would be grateful for references so we can properly acknowledge and discuss them.
>
>
> **Summary:**
>
> We would like to again thank the reviewer for the feedback; we believe that the revised paper will be stronger because of these suggestions and new experiments. We kindly ask that the reviewer reevaluate the paper in light of these revisions. We would be more than happy to run additional experiments, if there are any outstanding concerns about the paper.
>
> **References:**
>
> [1] Kraskov, A., et al., Phys. Rev. E, Estimating Mutual Information
>
> [2] Lehnert, L., et al., COLM 2024, Beyond A*: Better Planning with Transformers via Search Dynamics Bootstrapping
>
> [3] Bush, T., et al., ICLR 2025, Interpreting Emergent Planning in Model-Free Reinforcement Learning
>
> [4] Zawalski, M., et al., ICLR 2023, Fast and Precise: Adjusting Planning Horizon with Adaptive Subgoal Search

---

> > ### Author Response · Authors · 2025-08-04
> >
> > Dear Reviewer,
> >
> > We have carefully addressed your feedback through new experiments, additional analyses, and clarifications throughout the paper. We hope these improvements resolve your concerns and welcome any further thoughts you may have.
> >
> > Thanks again!
> >
> > The Authors

---

> > > ### Author Response · Authors · 2025-08-06
> > >
> > > Dear Reviewer,
> > >
> > > We want to share preliminary experimental results demonstrating that our method can be applied to solving SAT problems.
> > > In summary, we found that, even without any search, the distance function learned by contrastive learning was sufficient to solve 80% of the SAT instances, compared to only 10% when using a randomly initialized network. We provide more details below. We believe this extension would significantly increase the value and impact of our paper, and hope that the reviewer takes these new results into account when considering their final evaluation of the paper.
> > >
> > > Specifically, the experiment we ran investigated whether CRL can learn representations that enable inferring satisfying variable assignments in propositional formulas. We generated synthetic data by starting from a truth value ('t' or 'f') and applying transition rules such as 't' → '(t or f)'. After generating the full trajectory, we substituted each literal with an independent variable. This results in trajectories of the form: (x1 or (x2 and x3)) → (x1 or (x2 and t)) → (x1 or (f and t)) → (x1 or f) → (t or f) → t. We used 10 variables in total.
> > >
> > > To avoid trivial examples, we filtered the dataset to include only formulas that were not satisfied by any of 10 randomly sampled variable assignments.
> > >
> > > Additionally, inspecting the learned distance function revealed meaningful structure:
> > > expressions that evaluated to true (e.g., (t and (t or f))) were closer to 't' than to 'f', and vice versa. This suggests that the model captures the semantic content of the formulas.
> > >
> > > Although this is only a proof-of-concept experiment due to time constraints, these results suggest that CRL is capable of modeling the temporal structure present in SAT problems. If the paper is accepted, we will extend our results in the camera-ready version to include an evaluation on the G4SATBench benchmark [1],
> > > comparing our method to GNN-based baselines in the task of findings a satisfiability assignment.
> > >
> > > Best regards,
> > > The Authors
> > >
> > > [1] Zhaoyu, L., et al., Trans. Mach. Learn. Res. 2024, G4SATBench: Benchmarking and Advancing SAT Solving with Graph Neural Networks

---

> > > > ### Comment · Reviewer_BU6x · 2025-08-08
> > > >
> > > > I appreciate the author response and have read all the other reviews. Most of my gaps in understanding and concerns have been answered. I am able to appreciate the novelty regarding leveraging trajectory structure to construct informative negatives when using contrastive learning to solve combinatorial problems. The new experiments and analysis strengthen the paper. I have increased the score.

---

### Official Review · Reviewer_g7ur · 2025-07-02

**Clarity:** 4
**Significance:** 2
**Originality:** 2
**Rating:** 4
**Confidence:** 4

**Summary:**

Contrastive learning fails on combinatorial reasoning tasks because it learns to exploit superficial contextual features that disregard relevant temporal information (Fig. 1). The failure occurs when there's a constant "context" (like wall patterns in Sokoban) that makes it trivial to distinguish between trajectories. Standard contrastive learning samples negatives from different trajectories, so models can achieve perfect accuracy by identifying the context rather than learning meaningful temporal relationships (Fig. 5-6). To resolve this, CR^2 samples negative examples (temporally distant states) from within the same trajectory (in-trajectory negatives) rather than only from different trajectories.

CR^2 optimises $I(X; X_+) - I(X_+; C)$, encouraging representations that capture temporal structure while being invariant to context. Practically, this is done via a simple modification (Fig. 3) to how the data is loads; repeat trajectory indices in batches with a "repetition factor" (typically 2). Evaluation across 5 different settings (Sokoban, Rubik's Cube, N-Puzzle, Lights Out, Digit Jumper) shows that CR² consistently outperforms standard contrastive learning (Fig. 4) and matches supervised baselines using Best-First Search planning.

**Questions:**

1.	In the setting with partial observability, stochastic dynamics – Assumption 4.2 would not hold. How sensitive is the method to these violations?  Can we characterise exactly which MDP structures satisfy this assumption?
2.	Were other search algorithms evaluated apart from best-first search – that weren’t included in the manuscript? Example, A* ?
3.	We can see that the training objective minimises I(X⁺; C) i.e., mutual information between representations and context, should lead to context invariance. However, this is not shown empirically. Therefore, could we see how much the mutual information decreases during training for the different settings.

We can see that search is necessary to get appropriate solutions -- with certain environments requiring more of the search budget (6000 nodes).

4.	Can we quantify the relationship between representation quality (measured by correlation with true distances) and required search budget?
5.	Does a 10% improvement in correlation translate to a predictable reduction in nodes explored?
6.	How do these trade-offs change with problem size (15-puzzle vs. 24-puzzle)?
7.	Is this relationship linear, logarithmic, or domain-dependent?
8.	Is there a threshold correlation value below which search becomes ineffective?


9.	For Fig.8 would it be possible to include confidence intervals? this would allow a concrete evaluation of the CRL and CR^2 and importantly the robustness of the evaluation.
10.	The results show that having higher repetition factor degrade performance. Is this because the higher R is increasing too much noise in the training signal?
11.	Minor point: Last equality of equation 4 is repeated. It would be good to remove it?

**Ethical Concerns:**

["NO or VERY MINOR ethics concerns only"]

**Final Justification:**

All my concerns were addressed during the review process - I recommend acceptance.

**Limitations:**

Yes

**Quality:**

3

**Strengths And Weaknesses:**

Strength: Overall, I like this paper – it provides a nice example of why standard contrastive learning fail on combinatorial problems, proposes a simple fix using in-trajectory negatives and shows consistent improvements across multiple domains. Improves search efficiency when combined with Best-First Search planning. The code is made public for future extensions.

Limitation:  Despite the well-structured evaluations, there are some limitations. The problems considered are quite small-scale and evaluation on more realistic problem settings might be useful to evaluate the significance of the contribution. Some of the results presented seem to be for limited (i.e., 3) or single seeds, making it difficult to evaluate their robustness. The restrictive Assumption 4.2 (X_i ⊥ C | X_j) may not hold in many real-world domains with partial observability or stochastic dynamics, limiting applicability beyond clean combinatorial puzzles. Furthermore, the paper shows that despite the improvements in representation, CR^2 still cannot replace search entirely – particularly in limited budget settings. Table 1 shows solve rates near zero for complex problems without search (15-puzzle). This suggests we may be hitting fundamental limits of what these types of representations can achieve for combinatorial reasoning.

---

> ### Author Rebuttal · Authors · 2025-07-31
>
> We thank the reviewer for their detailed and constructive feedback.
>
> **> Were other search algorithms evaluated apart from best-first search – that weren’t included in the manuscript? Example, A\*?**
>
> Following the reviewer’s comment, we ran preliminary A* experiments on the Rubik’s Cube to test whether CR$^2$ representations remain effective beyond greedy search. Using a node budget of 6000 and varying $\alpha$ in the priority function $heuristic + \alpha \cdot cost$ for $\alpha \in {0, \dots, 400}$, CR$^2$ consistently outperforms CRL in both success rate and solution length. This suggests that CR$^2$ generalizes well across search strategies. We will clarify this point and include the results in the revised paper. The full results are in the table below:
> | $\alpha$              	| 0 	| 100   | 200   | 300   | 400   |
> |-----------------------------|-------|-------|-------|-------|-------|
> | **CR$^2$ Avg. Solution Length** | 56.76 | 46.35 | 38.42 | 32.84 | 29.16 |
> | **CR$^2$ Success Rate**     	| 0.63  | 0.62  | 0.59  | 0.54  | 0.33  |
> | **CRL Avg. Solution Length**	| 62.96 | 49.88 | 41.94 | 36.11 | 31.77 |
> | **CRL Success Rate**       	| 0.54  | 0.50  | 0.44  | 0.40  | 0.30  |
>
>
>
>
> **> We can see that the training objective minimises I(X⁺; C) i.e., mutual information between representations and context, should lead to context invariance. Therefore, could we see how much the mutual information decreases during training for the different settings.**
>
> We thank the reviewer for the suggestion. To empirically validate that the training objective minimizes $I(X^+; C)$ while maximizing $I(X; X^+ \mid C)$, we estimate mutual information using the NPEET package [3], which uses a k-nearest neighbor method. We analyze over 45k transitions from Sokoban and over 20k from Digit Jumper.
>
> The variables are defined as:
> - $X$: Z-score normalized current state embeddings,
> - $X^+$: Next state embeddings, standardized and projected using the same parameters as $X$,
> - $C$: Trajectory IDs, encoded as 2D samples from $\mathcal{N}(0, I_2)$.
>
> To mitigate the curse of dimensionality, we reduce all variables to low-dimensional spaces (3D for embeddings, 2D for $C$). The tables report $I(X; X^+ \mid C)$, $I(X; X^+)$, and $I(X^+; C)$ for CRL and CR$^2$ using $k = 3, 4, 5$ to show estimation stability.
> ## Sokoban
> ### CRL
> | MI type     |   $k=3$ |  $ k=4$ |  $ k=5 $|
> |-------------------|-------|-------|-------|
> | $I(X;X^+ \mid C)$ |  1.2  |  1.13 |  1.07 |
> | $I(X;X^+)$     | 11.01 | 10.93 | 10.87 |
> | $I(X^+;C)$     |  9.8  |  9.8  |  9.8  |
>
> ### CR$^2$
> | MI type     |   $k=3$ |  $ k=4$ |  $ k=5 $|
> |-------------------|-------|-------|-------|
> | $I(X;X^+ \mid C)$ |  6.47 |  6.56 |  6.6  |
> | $I(X;X^+)$     |  7.26 |  7.2  |  7.14 |
> | $I(X^+;C)$   |  0.79 |  0.64 |  0.55 |
>
> ## Digit Jumper
> ### CRL
> | MI type     |   $k=3$ |  $ k=4$ |  $ k=5 $|
> |-------------------|-------|-------|-------|
> | $I(X;X^+ \mid C)$ |  1.76 |  1.97 |  2.14 |
> |  $I(X;X^+)$     |  9.4  |  9.41 |  9.4  |
> | $I(X^+;C)$     |  7.64 |  7.44 |  7.26 |
>
> ### CR$^2$
> | MI type     |   $k=3$ |  $ k=4$ |  $ k=5 $|
> |-------------------|-------|-------|-------|
> | $I(X;X^+ \mid C)$ |  3.96 |  4.12 |  4.24 |
> |  $I(X;X^+)$     |  5.84 |  5.8  |  5.75 |
> |$I(X^+;C)$   |  1.89 |  1.68 |  1.51 |
>
> We observe that CR$^2$ indeed minimizes $I(X^+; C)$, while maximizing $I(X;X^+)$, which leads to the desired maximization of $I(X;X^+\mid C)$ for both environments. In contrast, CRL baseline results in high values of $I(X^+;C)$, which contribute to relatively low conditional mutual information $I(X;X^+\mid C)$.
>
> We have briefly described this analysis at the end of Section 4.2 (Learning Representations that Ignore Context) and put the detailed results in the Appendix.
>
> **> Can we quantify the relationship between representation quality (measured by correlation with true distances) and required search budget?**
>
> This is a highly relevant and insightful question. While a full analysis is ongoing, we investigated a closely related quantity: the relationship between correlation and the success rate at a fixed budget (6000 nodes).
>
> We consider three environments: Sokoban with 12x12 boards, Sokoban with 16x16 boards and the Rubik’s Cube. We run a grid of 96 short trainings for each one of them, varying the network architecture hyperparameters and the metric choice for contrastive learning.
>
> We find that for each one of them, a function of the form $\frac{1}{a(x - b)^2} +c$ describes this relationship well. The performance is almost constant up to some point and for values close to 1 grows very quickly.
>
> We find the optimal values to be:
> - Sokoban 12x12 $a=5000, b=0.05, c=0.05$ with MSE 0.02,
> - Sokoban 16x16 $a = 800, b = 0, c=0.05$ with MSE 0.14,
> - Rubik’s Cube $a=1000, b=0.02, c=0$ with MSE 0.02.
>
> Therefore, the solved rate is effectively 0 for correlation smaller than:
> - Sokoban 12x12: $0.8$,
> - Sokoban 16x16: $0.85$,
> - Rubik’s Cube: $0.9$.
>
>
> **> For Fig.8 would it be possible to include confidence intervals?**
>
> Thank you for the suggestion. During the rebuttal period, we re-ran the experiments from Figure 8 using 5 random seeds, varying both the train-test split and data order. The standard error was consistently low across tasks, with the exception of CRL on the Hammer and Pen tasks, where it reached approximately 0.05 still not affecting the overall conclusions. We will update the figure in the revised version to include confidence intervals.
>
>
>
> **> The results show that having higher repetition factor degrade performance. Is this because the higher R is increasing too much noise in the training signal?**
>
> Increased noise in the training signal is indeed a likely contributor to reduced performance. While more negative examples typically reduce gradient variance [1], a high repetition factor can counteract this by introducing highly similar negatives, limiting this benefit. Additionally, higher repetition may introduce bias, as observed in small-batch contrastive learning setups [2]. We have added this discussion to Section 5.6.
>
> **> Minor point: Last equality of equation 4 is repeated. It would be good to remove it?**
>
> Thank you for identifying this typo. It has been corrected in the revised version.
>
> **> The problems considered are quite small-scale and evaluation on more realistic problem settings might be useful to evaluate the significance of the contribution.**
>
> Thank you for the comment. We agree that evaluating on real-world tasks such as retrosynthesis or robotic assembly would further demonstrate the practical significance of our method, and we note this direction in the Limitations section. However, we would like to clarify that the problems we study are not small-scale. For instance, the Rubik’s Cube has a state space of $4.3 \times 10^{19}$, and to our knowledge, no existing method has learned high-quality representations in this domain. Successfully doing so highlights our method’s ability to model highly complex and nontrivial distributions.
>
> **> Some of the results presented seem to be for limited (i.e., 3) or single seeds, making it difficult to evaluate their robustness.**
>
> Thank you for pointing this out. Some of the original plots were indeed generated from a limited number of seeds. We are currently re-running all experiments with 5 random seeds to improve statistical robustness. Several runs have already completed, and the results closely match those previously reported. We will include updated figures with error bars in the camera-ready version.
>
> **> In the setting with partial observability, stochastic dynamics – Assumption 4.2 would not hold. How sensitive is the method to these violations? Can we characterise exactly which MDP structures satisfy this assumption?**
>
> Thank you for this insightful question. Assumption 4.2 requires the context to remain fixed and fully observable across a trajectory, which may not hold in partially observable environments. However, such settings are beyond our scope. We focus on fully observable, deterministic environments—an important and challenging class that includes combinatorial puzzles (e.g., Sokoban, 15-puzzle, Rubik’s Cube) and practical problems (e.g., Travelling Salesman, Robotic Assembly, Retrosynthesis), and an active area of research [3, 4, 5]. We have clarified this in the Limitations section.
>
> **> Furthermore, the paper shows that despite the improvements in representation. This suggests we may be hitting fundamental limits of what these types of representations can achieve for combinatorial reasoning.**
>
> Thank you for this observation. We agree that understanding the limits of learned representations is important and underexplored, and see our work as a step in that direction. While success rates on complex tasks like the 15-puzzle remain low under strict step budgets without search, Appendix E shows that relaxing episode length substantially improves solve rates. This suggests the limitation stems more from fixed planning horizons than a lack of generalization. Additionally, on the 15-puzzle, both CR$^2$ and CRL produce solutions shorter than those in the demonstration set, indicating meaningful generalization. CR$^2$ also significantly outperforms CRL on both the 15-puzzle and Rubik’s Cube, highlighting its strength even without search.
>
>
> **References:**
>
> [1] Cho, J., et al.,  TMLR 2024, Mini-Batch Optimization of Contrastive Loss
>
> [2] Chen. C., et al., NeurIPS 2022, Why do We Need Large Batchsizes in Contrastive Learning? A Gradient-Bias Perspective
>
> [3] Kraskov, A., et al., Phys. Rev. E, Estimating Mutual Information
>
> [4] Lehnert, L., et al., COLM 2024, Beyond A*: Better Planning with Transformers via Search Dynamics Bootstrapping
>
> [5] Bush, T., et al., ICLR 2025, Interpreting Emergent Planning in Model-Free Reinforcement Learning
>
> [6] Zawalski, M., et al., ICLR 2023, Fast and Precise: Adjusting Planning Horizon with Adaptive Subgoal Search

---

> > ### Author Response · Authors · 2025-08-04
> >
> > Dear Reviewer,
> >
> > Thank you again for your thoughtful feedback. In our response, we’ve added new experiments, analysis, and clarifications to address your comments. We hope the updates above resolve your concerns, and we’d be grateful for any further thoughts.
> >
> > Thanks!
> >
> > The Authors

---

> > ### Comment · Reviewer_g7ur · 2025-08-07
> >
> > Thank you for the thorough rebuttal and additional comments. They are helpful - I maintain my acceptance of the paper.
> >
> > One final comment - it would be beneficial to have a small section in the main paper to highlight the results from Appendix E. It would help situate the results appropriately.

---

> > > ### Author Response · Authors · 2025-08-07
> > >
> > > Dear Reviewer,
> > >
> > > Thank you! We'll make sure that the small section is included in the main paper. Please don’t hesitate to let us know if you have any further feedback or suggestions for improvement.
> > >
> > > Kind regards,
> > > The Authors

---

### Official Review · Reviewer_TTPY · 2025-07-03

**Clarity:** 4
**Significance:** 2
**Originality:** 2
**Rating:** 4
**Confidence:** 2

**Summary:**

The paper introduces CR2, a contrastive representation learning method for solving multi-step combinatorial problems. The paper demonstrates the failure of standard contrastive learning approaches in solving combinatorial problems due to the presence of excessive non-essential information, commonly referred to as "common context," in all instances of the same trajectory. Based on this observation, the paper goes on to propose a method to introduce in-trajectory negatives. This is achieved simply by replicating the trajectories a fixed number of times in the same batch and leaving the rest of the CL pipeline as is, which has the effect of randomly introducing negatives from the same trajectory. The experiments demonstrate that CR2 improves significantly over the naive CRL objective. It is also competitive with supervised value-based methods.

**Questions:**

1. Isn't repeating trajectories (as described in Figure 3) in a batch equivalent to randomly (perhaps not uniformly randomly) sampling in-trajectory "hard negatives"?

2. Why does the training objective saturate for CR2 in Figure 5? Is it to be expected due to in-trajectory negatives?

**Ethical Concerns:**

["NO or VERY MINOR ethics concerns only"]

**Final Justification:**

I maintain my score. The paper presents a novel idea that works, but the experiments, which only include tasks with deterministic dynamics and noiseless states, are a bit toyish.

**Limitations:**

yes

**Quality:**

3

**Strengths And Weaknesses:**

**Quality**: All claims made in the paper are well-supported by targeted experiments. The limitations of the current work, i.e., the use of deterministic known dynamics and noiseless states, are clearly stated.

**Clarity**: The paper is well written and easy to follow. The intuition provided using probabilistic arguments is helpful in understanding the general principle behind the method in a task-agnostic manner.

**Significance**: Using deterministic dynamics and noiseless states makes the experimental setup seem somewhat toy-like. The paper's impact could be greatly enhanced if experiments on more realistic tasks were also included.

**Originality**: Although the idea in the paper closely relates to the concept of "hard negatives" in general contrastive representation learning literature, I believe the way the idea is presented and instantiated is original.

---

> ### Author Rebuttal · Authors · 2025-07-31
>
> We thank the reviewer for reading our paper and providing feedback!
>
> **> Isn't repeating trajectories (as described in Figure 3) in a batch equivalent to randomly (perhaps not uniformly randomly) sampling in-trajectory "hard negatives"?**
>
> Thank you for the thoughtful question. While at first glance repeating trajectories in a batch may seem equivalent to sampling in-trajectory hard negatives, the two approaches behave differently. In standard contrastive learning (as in CRL), an anchor $x$ pulls its positive $x_+$ closer and pushes negatives (e.g., $y_+$) away. However, negatives like $y_+$ are simultaneously pulled by their own anchors (e.g., $y$), which limits how far they are pushed by $x$.
> In contrast, when using in-trajectory negatives without anchoring them (e.g., $x$ pushes $x_-$ away, but $x_-$ has no anchor), these states can drift arbitrarily far in representation space. This is problematic, especially since in-trajectory negatives are harder (closer in structure), which results in stronger gradient updates.
> Our proposed method, $CR^2$, addresses this by anchoring all in-trajectory negatives. This keeps trajectories coherent and prevents such drift. Empirical evidence supporting this can be found in Appendix G, Figures 23 and 24.
> In the camera-ready version, we will include this discussion in section 4.3 (Method used in practice)
>
> **> Why does the training objective saturate for CR2 in Figure 5? Is it to be expected due to in-trajectory negatives?**
>
> Yes, this saturation is expected and arises due to the use of in-trajectory negatives. Figure 5 shows, for each anchor, how often the closest state is its true positive. With in-trajectory negatives, it’s possible for a negative from the same trajectory to be embedded closer than the positive. Additionally, once wall-based features are no longer sufficient for classification, the model may start relying on more abstract or subtle features. As a result, positives from other trajectories may occasionally be embedded closer to an anchor, contributing to the observed saturation.
> We have included this explanation in Section 5.3, where the Figure 5 is described.
>
> **> Using deterministic dynamics and noiseless states makes the experimental setup seem somewhat toy-like. The paper's impact could be greatly enhanced if experiments on more realistic tasks were also included.**
>
> Thank you for this suggestion. We agree that extending the method to more realistic tasks is an important direction. In this work, we focused on deterministic dynamics and noiseless states, as these are standard in combinatorial puzzles. As discussed in the Limitations section, we plan to extend our approach to real-world combinatorial problems, such as robotic assembly and retrosynthesis. If the reviewer has specific suggestions for additional practical combinatorial domains, we would be happy to include them in the Limitations section for the camera-ready version and explore them in future work.

---

> > ### Author Response · Authors · 2025-08-04
> >
> > Dear Reviewer,
> >
> > We thank you again for your feedback! We’ve addressed your comments by adding clarifications. Do the responses above resolve your concerns? We would greatly appreciate your engagement.
> >
> > Thanks!
> >
> > The Authors

---

> > > ### Comment · Reviewer_TTPY · 2025-08-04
> > > **Thanks for answering my questions**
> > >
> > > Thanks for answering my questions. It helps me understand the key novelty of the work better. Regarding the question about some realistic tasks, two come to mind: simple zebra puzzles (perhaps some synthetic form) and Sudoku puzzles. These two tasks can be expressed as sequence generation and have some flavor of search.

---

> > > > ### Author Response · Authors · 2025-08-07
> > > >
> > > > Dear Reviewer,
> > > >
> > > > Thank you for the confirmation and for suggesting additional tasks we could explore. As you noted, both zebra puzzles and Sudoku can be naturally framed as sequence generation problems with a search component, making them a promising fit for our approach.
> > > >
> > > > If the paper is accepted, we will aim to include these experiments in the camera-ready version.
> > > >
> > > > Best regards,
> > > > The Authors

---

### Official Review · Reviewer_3TVa · 2025-07-06

**Clarity:** 4
**Significance:** 4
**Originality:** 3
**Rating:** 5
**Confidence:** 3

**Summary:**

The paper presents CR$^2$ (Contrastive Representations for Combinatorial Reasoning), which extends temporal contrastive learning with in-trajectory negatives to learn embeddings that double as effective search heuristics.
By revealing how standard contrastive RL collapses onto static context and then maximizing conditional mutual information via a simple “repetition factor,” the method preserves the temporal signal crucial for planning. Evaluated on several tasks like complex puzzles and control problems, CR2’s embeddings align closely with true state-space distances, lifting correlation metrics and solve rates and, in many settings, matching the performance of heavyweight search-centric systems. However, further analysis confirms that explicit search is still required on the harder instances.

**Questions:**

None

**Ethical Concerns:**

["NO or VERY MINOR ethics concerns only"]

**Limitations:**

Yes

**Paper Formatting Concerns:**

Line 210 -- Instead of Algorithm 4.3, it should say Figure 3.

**Quality:**

4

**Strengths And Weaknesses:**

**Strenghts**

- The paper is well written and easy to follow.
- The paper identifies an important flaw with standard contrastive RL approaches, especially for combinatorial reasoning tasks, and provides a solution -- using CR$^2$ that learns to ignore the context and focus on the temporal and causal structure.
- The paper provides further analysis and experiments where they investigate CR$^2$ such as testing if search is still needed, experimenting in cases where the context is dynamic.

**Weaknesses**

- The conclusions from the t-SNE plots are not very clear, and the claim that the plots for CRL form clusters, indicating they learn the context (walls), while CR$^2$ seems to be a bit hand-wavy. Perhaps this argument could be strengthened.


Overall, I think the paper makes a good contribution, and I recommend acceptance

---

> ### Author Rebuttal · Authors · 2025-07-31
>
> We thank the Reviewer for their effort to assess our work! The main concern raised appears to be that our conclusions drawn from the t-SNE plots require stronger empirical support. In order to do that we propose two additional experiments, one based on training a linear classifier on top of our representations and another one using a mutual information estimator, based on our learned representations. The details are contained in the response below.
>
> **> The conclusions from the t-SNE plots are not very clear, and the claim that the plots for CRL form clusters, indicating they learn the context (walls), while CR$^2$ does not, seems to be a bit hand-wavy. Perhaps this argument could be strengthened.**
>
> To support the conclusions drawn from the t-SNE plots, we conducted two experiments to assess how much information about the context do the CRL and CR$^2$ representations contain.
>
> # Experiment 1
> The experimental setup is simple: we train a linear classifier using representions from CRL, CR$^2$, or a randomly initialized network as input and use the trajectory ID as the prediction target. The trajectory ID serves as a proxy for the static context (e.g., wall configuration), since each trajectory is tied to a fixed environment layout. We use Sokoban trajectories (>49k transitions), with 70% of transitions from each trajectory used for training and the remaining 30% for testing. To obtain reliable accuracy estimates, we perform 6-fold cross-validation.
>
> Intuitively, if the representations contain contextual information, it should be easier for the classifier to predict the correct trajectory ID.
>
> The results are as follows (accuracy mean ± standard deviation):
> - CRL: 88.16 ± 0.30%,
> - CR$^2$: 1.80 ± 0.09%,
> - Randomly initialized representations: 3.32 ± 0.15%.
>
> The results show that trajectory ID—which reflects contextual information—can be accurately predicted from CRL representations in Sokoban. CR$^2$ representations achieve accuracy significantly lower than a random baseline, indicating that contextual signals are effectively suppressed.
>
> # Experiment 2
> In the following experiment, we estimate the mentioned mutual information using [NPEET package](https://github.com/gregversteeg/NPEET), which implements the method proposed in [1] that uses k-nearest neighbours to get entropy estimations. We conduct the analysis using trajectories collected from the Sokoban or Digit Jumper environment, utilizing all transitions within these trajectories (>45k transitions for Sokoban and >20k for Digit Jumper). The variables used in the experiment are defined as follows:
> - $X$: Current state embeddings, standardized using z-score normalization (mean 0, standard deviation 1) across the dataset. These embeddings are then projected onto a 3-dimensional subspace using Principal Component Analysis (PCA).
> - $X^+$: Next state embeddings corresponding to transitions from $X$. The same standardization parameters and PCA transformation applied to $X$ are used for $X^+$ to ensure consistency.
> - $C$: Trajectory identifiers (traj_id) encoded as 2-dimensional vectors sampled from a standard bivariate Gaussian distribution (i.e., $\mathcal{N}(0, I_2)$).
>
> Note: To mitigate the effects of the curse of dimensionality and ensure reliable performance of k-nearest neighbor (kNN)-based estimators, we reduce all high-dimensional representations to low-dimensional spaces (3D for state embeddings, 2D for trajectory identifiers).
> In the tables below, we report $I(X;X^+ \mid C)$, $I(X;X^+)$, and $I(X^+;C)$ for both CRL and CR$^2$ using 3 different numbers of neighbours ($k=3,4,5$) used for mutual information estimation. We report 3 different values of $k$ to demonstrate the stability of the estimation.
> ## Sokoban
> ### CRL Mutual Information (MI)
> | MI type     |   $k=3$ |  $ k=4$ |  $ k=5 $|
> |-------------------|-------|-------|-------|
> | $I(X;X^+ \mid C)$ |  1.2  |  1.13 |  1.07 |
> | $I(X;X^+)$     | 11.01 | 10.93 | 10.87 |
> | $I(X^+;C)$     |  9.8  |  9.8  |  9.8  |
>
> ### CR$^2$ Mutual Information (MI)
> | MI type     |   $k=3$ |  $ k=4$ |  $ k=5 $|
> |-------------------|-------|-------|-------|
> | $I(X;X^+ \mid C)$ |  6.47 |  6.56 |  6.6  |
> | $I(X;X^+)$     |  7.26 |  7.2  |  7.14 |
> | $I(X^+;C)$   |  0.79 |  0.64 |  0.55 |
>
> ## Digit Jumper
> ### CRL Mutual Information (MI)
> | MI type     |   $k=3$ |  $ k=4$ |  $ k=5 $|
> |-------------------|-------|-------|-------|
> | $I(X;X^+ \mid C)$ |  1.76 |  1.97 |  2.14 |
> |  $I(X;X^+)$     |  9.4  |  9.41 |  9.4  |
> | $I(X^+;C)$     |  7.64 |  7.44 |  7.26 |
>
> ### CR$^2$ Mutual Information (MI)
> | MI type     |   $k=3$ |  $ k=4$ |  $ k=5 $|
> |-------------------|-------|-------|-------|
> | $I(X;X^+ \mid C)$ |  3.96 |  4.12 |  4.24 |
> |  $I(X;X^+)$     |  5.84 |  5.8  |  5.75 |
> |$I(X^+;C)$   |  1.89 |  1.68 |  1.51 |
>
> We observe that CR$^2$ indeed minimizes $I(X^+; C)$, while maximizing $I(X;X^+)$, which leads to the desired maximization of $I(X;X^+\mid C)$ for both environments. In contrast, CRL baseline results in high values of $I(X^+;C)$, which contribute to relatively low conditional mutual information $I(X;X^+\mid C)$.
>
> We have briefly described this analysis at the end of Section 4.2 (Learning Representations that Ignore Context) and put the detailed results in the Appendix.
>
> **> Line 210 -- Instead of Algorithm 4.3, it should say Figure 3.**
>
> Thank you for catching this. We have corrected the reference on Line 210 to "Figure 3" as suggested.
>
> **Summary:**
>
> We hope these additions meaningfully strengthen the paper and address the reviewer’s concerns. We would be happy to engage further or provide additional experiments if needed.
>
> **References:**
>
> [1] Kraskov, A., et al., Phys. Rev. E, Estimating Mutual Information

---

> > ### Author Response · Authors · 2025-08-04
> >
> > Dear Reviewer,
> >
> > We have worked hard to incorporate the review feedback by running new experiments and revising the paper. Do the revisions and discussions above address your concerns? We would greatly appreciate your engagement.
> >
> > Thanks!
> >
> > The Authors

---

> > > ### Comment · Reviewer_3TVa · 2025-08-05
> > >
> > > Thanks for the response and additional experiments — they do strengthen the paper. I continue to recommend acceptance.

---

### Note · Authors · 2025-08-12

We thank the reviewers for their thoughtful evaluation. We're pleased that our paper was found to be well-written and clear (3TVa, TTPY), to address a key limitation in contrastive RL for combinatorial reasoning with CR$^2$ (3TVa, g7ur), and to provide strong experimental validation with consistent improvements (3TVa, TTPY, g7ur).

The reviewers' feedback helped us run several important experiments during the rebuttal period that strengthen our paper:

- **Solver robustness** (g7ur, BU6x): We ran preliminary tests with A* in addition to BestFS and found the same improvements over CRL. This suggests our method works across different solvers, not just BestFS.

- **Direct validation of our theoretical framework** (3TVa, g7ur, BU6x): Two experiments confirm that CR$^2$ actually removes context information as intended:

    - We trained classifiers to predict context from state representations. With CRL representations, the classifier achieved 88% accuracy. With CR² representations, accuracy dropped to random chance levels—the same as using a randomly initialized network.
    - We measured mutual information directly and found that CR$^2$ minimizes $I(X^+; C)$ while maximizing $I(X^+; X)$, exactly as our theory predicts. CRL only maximizes $I(X^+; X)$.

- **Representation quality metric validation** (g7ur): We found a predictable relationship between Spearman rank correlation and solving success rate in both Rubik's Cube and Sokoban. This demonstrates that the correlation serves as a reliable measure of representation quality that translates to actual performance across domains.

- **SAT extension** (BU6x): We ran preliminary experiments showing that contrastive representations can tackle SAT problems, extending beyond the combinatorial puzzle domains in our main experiments.

We also provided clarifications on the points raised by reviewers. These experiments and clarifications address the main concerns while strengthening our paper's empirical foundation.

We appreciate the reviewers' thorough feedback and believe these additions make a stronger case for CR$^2$ as an effective approach to contrastive learning in combinatorial reasoning.

---

### Decision · Program_Chairs · 2025-09-17

**Decision:**

Accept (poster)

**Comment:**

This paper proposes a contrastive learning method that uses in-trajectory negatives to learn representations for combinatorial reasoning tasks, which overcomes a failure mode of standard contrastive approaches.

The reviewers generally agreed that the paper is well-written and nicely motivated the problem by identifying an important limitation in existing methods. They also agreed that the proposed solution is simple yet effective, and the experimental results are convincing. There were are a few initial concerns about the novelty of the proposed method, the limited scope of the evaluation, and a need for more rigorous validation of the theoretical claims about context removal. However, the authors provided additional results from new experiments during the rebuttal period that directly addressed most of the concerns. As a result, a few reviewers increased their scores.

Given that the the paper got stronger over the rebuttal period, and that the reviewers are unanimously positive about the paper, I recommend acceptance.